# Features of repertoire diversity and gene expression in human cytotoxic T cells following allogeneic hematopoietic cell transplantation

Hideki Nakasone [1][✉], Machiko Kusuda[1], Kiriko Terasako-Saito[1], Koji Kawamura[1], Yu Akahoshi [1], Masakatsu Kawamura[1], Junko Takeshita[1], Shunto Kawamura[1], Nozomu Yoshino[1], Kazuki Yoshimura[1], Yukiko Misaki[1], Ayumi Gomyo[1], Kazuaki Kameda[1], Masaharu Tamaki[1], Aki Tanihara[1], Shun-ichi Kimura[1], Shinichi Kako[1] & Yoshinobu Kanda [1][✉]

Cytomegalovirus reactivation is still a critical concern following allogeneic hematopoietic cell transplantation, and cellular immune reconstitution of cytomegalovirus-specific cytotoxic T-cells is necessary for the long-term control of cytomegalovirus reactivation after allogeneic hematopoietic cell transplantation. Here we show the features of repertoire diversity and the gene expression profile of HLA-A24 cytomegalovirus-specific cytotoxic T-cells in actual recipients according to the cytomegalovirus reactivation pattern. A skewed preference for BV7 genes and sequential "G" amino acids motif is observed in complementarity-determining region-3 of T cell receptor-β. Increased binding scores are observed in T-cell clones with complementarity-determining region-3 of T cell receptor-β with a "(G)GG" motif. Single-cell RNA-sequence analyses demonstrate the homogenous distribution of the gene expression profile in individual cytomegalovirus-specific cytotoxic T-cells within each recipient. On the other hand, bulk RNA-sequence analyses reveal that gene expression profiles among patients are different according to the cytomegalovirus reactivation pattern, and are associated with cytokine production or cell division. These methods and results can help us to better understand immune reconstitution following hematopoietic cell transplantation, leading to future studies on the clinical application of adoptive T-cell therapies.

[1] Division of Hematology, Jichi Medical University Saitama Medical Center, Saitama, Japan. [✉]email: nakasone-tky@umin.ac.jp; ycanda-tky@umin.ac.jp

Cytomegalovirus (CMV) reactivation is still a critical concern following allogeneic hematopoietic cell transplantation (allo-HCT) or organ transplantation, since CMV diseases such as pneumonia are associated with high mortality[1–5]. Routine monitoring and the preemptive use of ganciclovir have been established to control CMV reactivation for recipients of allo-HCT[6–8]. However, cellular immune reconstitution of CMV-specific cytotoxic T-cells (CMV-CTL) is generally considered to be necessary for the long-term control of CMV reactivation after allo-HCT.

The identification of CMV-CTL clones is one method used to assess cellular immunity against CMV. We previously established a direct single-cell analysis to simultaneously identify and quantify in vivo CMV-CTL clones after allo-HCT[9,10]. However, that method required substantial time and effort. The recent development of next-generation sequence (NGS) technology has contributed to the high-throughput quantitative assessment of immune reconstitution following allo-HCT. An individual T-cell has a specific complementarity-determining region 3 (CDR3) of T cell receptor (TCR)-α and -β, which is a result of the recombination of somatic TCR V-(D)-J genes and junction diversity. NGS enables us to track these individual clones among ~million cells[11,12]. In fact, immune diversity and tracking of T-cells or B-cells following allo-HCT have been shown to be partially associated with clinical outcomes[13–17]. However, these reports have focused on the entire non-specific immune diversity, and not directly on the cellular immunity specific to virus.

Furthermore, an adoptive T-cell therapy has recently been reported as a possible promising treatment for severe CMV diseases as an alternative to antiviral agents, and several studies have actually proven the efficacy of these strategies[18–25]. A better and deeper understanding of immune reconstitution following allo-HCT is required for further clinical application of these adoptive T-cell therapies. However, little information is available on the gene expression profile (GEP) of CMV-CTL clones according to CMV reactivation patterns (and donor CMV serostatus) after allo-HCT.

Thus, we conducted immune- and RNA-sequencing of HLA-A24-restricted CMVpp65-specific CTLs to better understand the immune reconstitution of CMV-CTLs after allo-HCT. To the best of our knowledge, this is the first report on the features of TCRβ-CDR3, diversity, and GEP of HLA-A24 CMV-CTLs according to the CMV-reactivation pattern among recipients after allo-HCT. In addition, we further sought to demonstrate homogeneity or heterogeneity according to individual CTL clones using single-cell RNA-sequencing technology.

## Results

**Patient characteristics**. We analyzed TCR of CMV-CTLs in 51 samples obtained from 26 patients and 3 donors (Table 1), including 2 samples analyzed by the direct single-cell method alone and reported previously[9,10]: 26 samples in the early phase (1–3 months after allo-HCT), and 22 samples in the late phase (6 months to >1 year after allo-HCT). All of the recipients and their corresponding donors had to have HLA-A24:02 or -A24:20. The median ages of the recipients and donors were 45 (range:16–4) and 37 years (range:11–53), respectively. All recipients tested positive for CMV, while half of the corresponding donors tested negative for CMV. Of them, CMV reactivation was not observed in 10 (the no-CMV reactivation group). Only one episode of CMV reactivation was observed in 9 (the one-episode group), while 7 recipients experienced repeated CMV reactivation (the repeated CMV reactivation group). As expected, the CMV reactivation group frequently experienced grade 2-4 acute graft-versus-host disease (Table 1).

**Correlation between the direct single-cell and NGS methods**. First, we checked the consistency of the identified CMV-CTL clones between the current NGS strategy and our previous direct single-cell method[9,10] ($n = 5$ samples) (Supplementary Fig. 1a). The two methods showed a good correlation in clone proportions (Supplementary Fig. 1b, coefficient = 0.97, $P < 0.001$). We subsequently adopted the NGS strategy as an alternative to the direct single-cell method.

**Features of TCRβ-CDR3 in HLA-A24-restricted CMVpp65-specific CTL clones**. In total, TCRβ-CDR3 were analyzed in 19765 CMV-CTLs from the 51 samples [median: 215 cells (range: 30–1000)/sample], and 354 clones were identified (Supplementary Data 1). Of the clones, 347 were observed after allo-HCT and the remaining 7 clones were observed only in the donors.

A skewed preference was observed for the selection of BV and BJ families in TCRβ-CDR3 of HLA-A24 CMV-CTL clones ($P < 0.001$ by chi-square test). Especially, BV7 was the most frequently used (33%), and half of CMV-CTL clones with BV7 selected BJ2-1 or BJ2-3 (Fig. 1a).

The length of amino acids (AA) in TCRβ-CDR3 of all CMV-CTL clones seemed to be symmetrically distributed (Fig. 1b), and the most frequently observed AA length was 15 followed by 14. Frequently-selected AA motifs in their TCRβ-CDR3 are shown according to AA length and the CMV-reactivation pattern in Fig. 1c. Apparently, "G" tended to be preferentially selected at positions 7–11. Sequential "G" usage was frequently observed, especially in the no-CMV reactivation or one-episode groups (Fig. 1c). If we focused on the top dominant clones in individual recipients at the early or late phase, "G" was preferentially selected in positions 7–10 in TCRβ-CDR3 with 15 AA, while "G" or "T" was preferred for that with 14 AA (Supplementary Fig. 2a).

Next, we focused only on the major CMV-CTL clones that accounted for >5% of all CMV-CTLs among individual recipients in the early or late phases after allo-HCT. The distributions of AA length in TCRβ-CDR3 were different according to the CMV reactivation patterns (Fig. 2a). CMV-CTL clones more frequently recruited 17 AA of TCRβ-CDR3 in the no-CMV reactivation group, while those in the repeated CMV reactivation group frequently selected 14 AA. The AA length of TCRβ-CDR3 decreased in the following order: no-CMV reactivation > one-episode > repeated CMV reactivation groups ($P = 0.015$ by Jonckheere–Terpstra test for decreasing tendency). The major CMV-CTLs with 17 AA distinctly selected the "(G)#GG~" motif, and those with 15 AA preferred the "GGG" motif (Supplementary Fig. 2b). If we consider the selected AA of TCRβ-CDR3 according to the CMV-reactivation pattern, the "GGG" motif in CMV-CTLs with 15 AA seemed to still be frequently selected among the major clones in the no-CMV reactivation group and partially in the one-episode group, but no common motif was observed in the repeated CMV reactivation group (Fig. 2b). Regarding CMV-CTLs with some other AA length, there were too few to create AA sequence logos according to the CMV-reactivation pattern. There was no difference in AA length of TCR ($P = 0.22$) or the presence of the "GGG" motif between CMV-seropositive and -seronegative donor groups ($P = 0.56$).

**TCR-peptide binding score of TCRβ-CDR3 of CMV-CTLs**. Based on the results described above, we hypothesized that CMV-CTL clones with "GGG" in ≤15 AA or "GG" in ≥16 AA of their TCRβ-CDR3 ["(G)GG" motifs] might demonstrate an increased binding affinity toward the peptide, or that the clones in the no-CMV reactivation group might, on average, show a high binding affinity. When we focused on the major clones that accounted for >5% of all CMV-CTLs among individual recipients in the early or

**Table 1 Patient characteristics.**

| | | [a]No CMV reactivation | One episode | Repeated CMV reactivation | *P*-value |
|---|---|---|---|---|---|
| Age | Median (range) | 43 (16–62) | 44 (26–64) | 45 (22–61) | 0.60 |
| Donor age | Median (range) | 39.5 (11–53) | 34 (23–50) | 45 (12–51) | 0.88 |
| Patient sex | Female | 4 | 1 | 1 | 0.37 |
| | Male | 6 | 8 | 6 | |
| Donor sex | Female | 3 | 5 | 4 | 0.43 |
| | Male | 7 | 4 | 3 | |
| Disease | AML | 7 | 6 | 1 | 0.08 |
| | ALL | 2 | 3 | 2 | |
| | MPN/MDS | 1 | 0 | 2 | |
| | ML/Other | 0 | 0 | 2 | |
| CMV serostatus | Donor−/recipient + | 5 | 3 | 5 | 0.66 |
| | Donor+/recipient + | 3 | 4 | 2 | |
| Disease status | Standard | 5 | 6 | 4 | 0.88 |
| | Advanced | 5 | 3 | 3 | |
| Donor | MRD-PB | 4 | 1 | 2 | 0.46 |
| | MMRD-PB[b] | 1 | 0 | 1 | |
| | MUD-BM | 4 | 4 | 1 | |
| | MUD-PB | 0 | 1 | 0 | |
| | MMUD-BM[b] | 1 | 3 | 3 | |
| Conditioning | MAC | 10 | 8 | 5 | 0.17 |
| | RIC | 0 | 1 | 2 | |
| GVHD prophylaxis | CsA-based | 7 | 7 | 6 | 0.85 |
| | Tac-based | 3 | 2 | 1 | |
| In vivo T-cell depletion | No | 9 | 8 | 6 | 1.00 |
| | Yes | 1 | 1 | 1 | |
| Grade2–4 aGVHD | No | 10 | 8 | 2 | 0.0012 |
| | Yes | 0 | 1 | 5 | |

*AML* acute myelogeneous leukemia, *ALL* acute lymphoblastic leukemia, *MDS* myelodysplastic syndrome, *MPN* myeloproliferative neoplasm, *ML* malignant lymphoma, *CMV* cytomegalovirus, *MRD* HLA-matched related donor, *MMRD* HLA-mismatched related donor, *MUD* HLA-matched unrelated donor, *MMUD* HLA-mismatched unrelated donor, *PB* peripheral blood stem cell, *BM* bone marrow, *MAC* myeloablative conditioning, *RIC* reduced-intensity conditioning, *GVHD* graft-versus-host disease, *CsA* cyclosporine, *TAC* tacrolimus, *aGVHD* acute GVHD.
[a]CMV reactivation was defined when ≥3 CMV antigenemia were detected.
[b]All of the recipients and their corresponding donors shared HLA-A24:02 or -A24:20.

late phases after allo-HCT, the estimated TCR-peptide binding score by ERGO algorithms[26] significantly differed according to the AA length of TCRβ-CDR3 (Fig. 2c). The binding score tended to be higher in the CMV-seropositive donor groups ($P = 0.06$). In addition, the scores were significantly higher in CMV-CTL clones with "(G)GG" motifs (Fig. 2d). This tendency was also confirmed in the analyses of all clones, and "(G)GG" motifs were frequently observed in CMV-CTLs with higher binding scores (Supplementary Fig. 3a–c). Clones with a higher binding score equally appeared in the repeated CMV reactivation group as well as in the other groups (Fig. 2e). However, when their binding scores were weighted by the estimated cell counts of individual clones in cryopreserved PBMCs isolated from 10 ml of peripheral blood, the weighted values were higher in the no-reactivation group in the early phase after allo-HCT among the major clones ($P = 0.02$ by the Kruskal–Wallis test and $P = 0.031$ by the J–T test for decreasing tendency, Fig. 2f). On the other hand, the weighted values were not significantly different according to the CMV reactivation group in the late phase after allo-HCT ($P = 0.13$ by the Kruskal–Wallis test).

**Changes in the proportion and diversity of HLA-A24-restricted CMVpp65-specific CTLs.** The proportion of CMV-CTLs to CD8[+]T-cells at the early phase of allo-HCT was not associated with their Shannon's equitability (the normalized Shannon's diversity index, where a lower value indicates less evenness and the presence of a dominant clone)(Supplementary Fig. 4a). In addition, no decreasing or increasing tendency was observed in the proportion or diversity of CMV-CTLs according to the CMV-reactivation pattern (Supplementary Fig. 4b and c). When we considered the estimated cell counts of CMV-CTLs in

10 ml of peripheral blood, the median cell count in the repeated CMV reactivation group was significantly lower than those in the other groups: 153 cells (range: 86–1030 cells) vs. 1760 cells (range: 60–3000 cells) in the one-episode group vs. 880 cells (range: 142–3000 cells) in the no-CMV reactivation group, $P = 0.046$), suggesting that the actual CMV-CTL counts would be critical for CMV control after allo-HCT. Focusing on time-dependent changes (Fig. 3a), the proportions of CMV-CTLs decreased in the late phases ($P < 0.01$, Fig. 3b) and their diversity increased ($P = 0.046$, Fig. 3c) when all groups were pooled. However, the statistical significance tended to diminish if we checked the time-dependent changes according to the CMV reactivation pattern (Supplementary Fig. 4d and e). If we consider the estimated cell counts, there were no significant time-dependent changes according to the CMV reactivation pattern.

If we focus on the three pairs of donor and recipient in the no-CMV reactivation group transplanted from CMV-seropositive donors, the dominant clones of each donor remained dominant after allo-HCT (Supplementary Fig. 4f), suggesting that CMV-CTLs transferred from the donors kept playing an anti-viral role in the cohort. Similarly, if we consider the most dominant top 1 clones of individual patients in all groups, there was no significant time-dependent proportional change according to the CMV-reactivation pattern between the early and late phases after allo-HCT (Supplementary Fig. 4g). This meant that the dominant clone in the early phases after allo-HCT usually remained dominant. However, in several cases, some dominant clones decreased in the late phases after allo-HCT (Fig. 3a). Next, we focused on the dominant clones that accounted for >35% of all CMV-CTLs within individual recipients in the early or late phases after allo-HCT, and checked the difference in TCR-binding scores

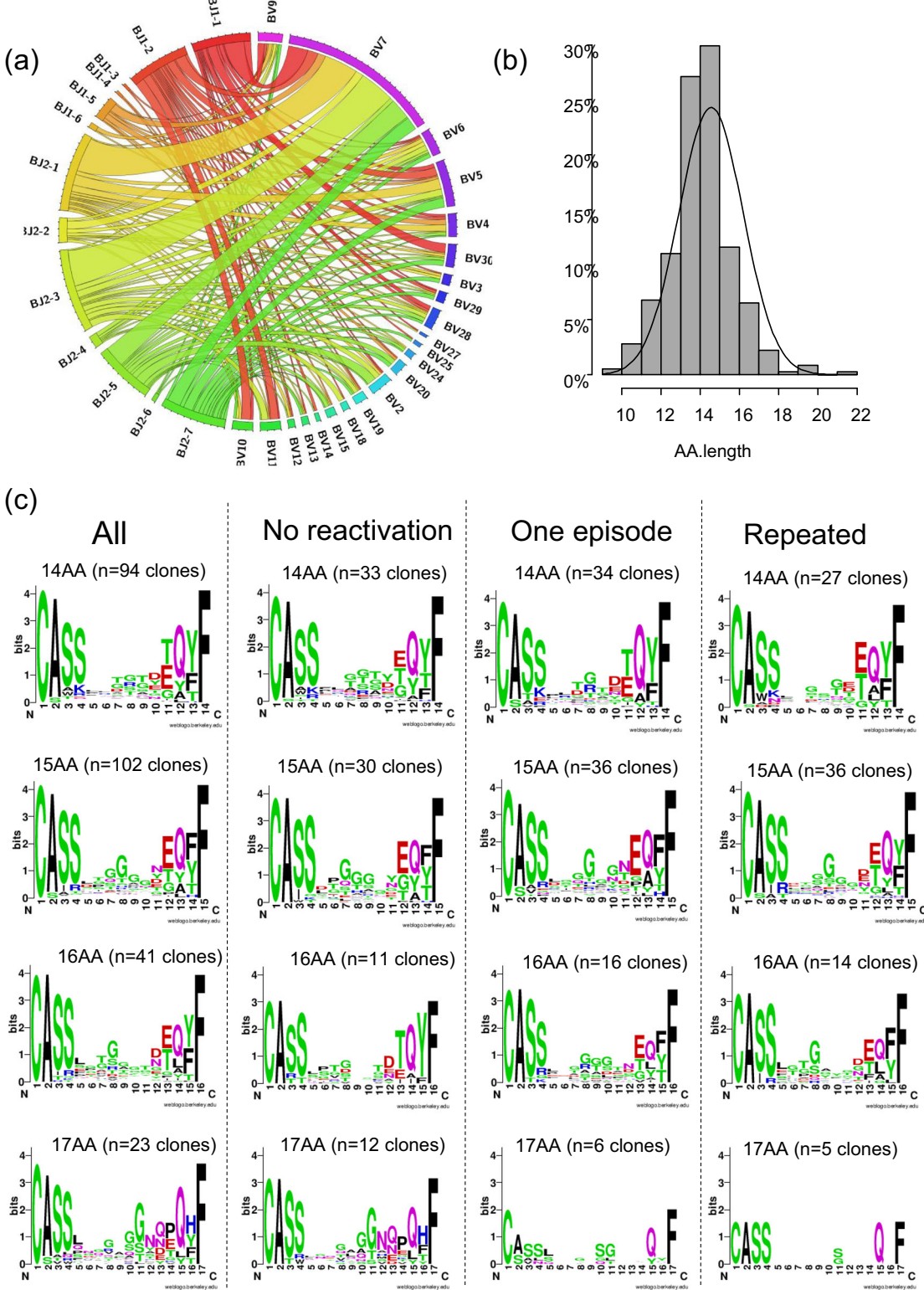

**Fig. 1 Preference in the selection of complementarity-determining region 3 (CDR3) of T cell receptor (TCR)-β of all identified HLA-A24-restricted CMV-pp65-specific cytotoxic T-cell (CMV-CTL) clones. a** A circos plot showing a skewed preference in the selection of BV and BJ genes of all CMV-CTL clones. **b** A column chart for the amino acid (AA) length of all CMV-CTL clones. **c** AA sequence logos in all CMV-CTLs observed after allo-HCT with 17, 16, 15, or 14 AA-long TCRβ-CDR3, and logos in the groups divided according to AA length and CMV reactivation pattern ($n = 10$ in the no-CMV reactivation group, $n = 9$ in the one-episode group, and $n = 7$ in the repeated-CMV reactivation group). X-axis denotes the position of amino acids from the N- to C-terminal.

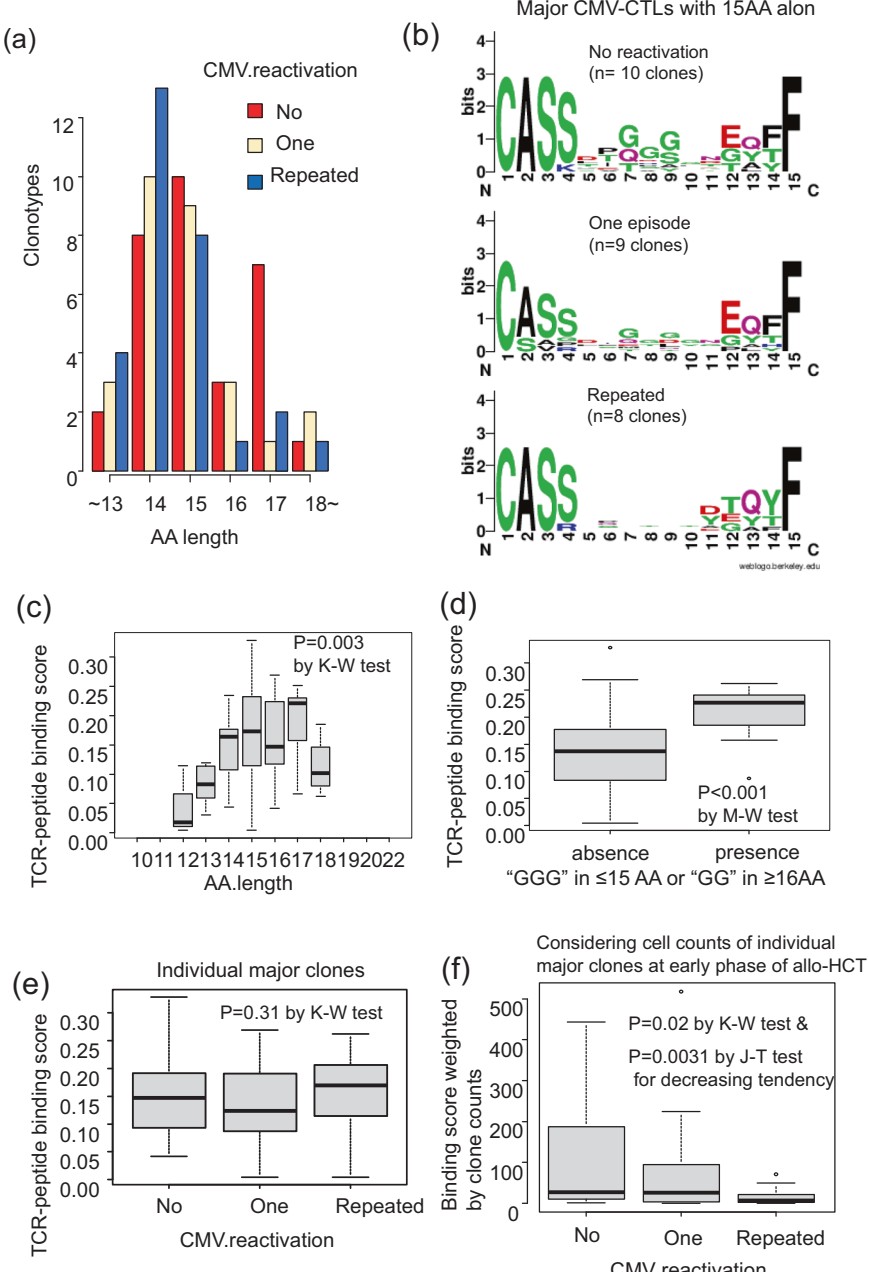

**Fig. 2 Amino acid (AA) sequence logos and AA length in the major HLA-A24-restricted CMV-pp65-specific cytotoxic T-cell (CMV-CTL) clones accounting for >5% of all CMV-CTLs within individual recipients in the early or late phases after allo-HCT. a** AA length according to CMV-reactivation pattern: no-CMV reactivation ($n = 10$), one-episode ($n = 9$), and repeated reactivation ($n = 7$) groups. The AA length of TCRβ-CDR3 decreased in the following order: no-CMV reactivation > one-episode ($n = 9$) > repeated CMV reactivation groups ($P = 0.015$ by the Jonckheere–Terpstra (J–T) test for decreasing tendency). **b** AA sequence logos of TCRβ-CDR3 in the major CMV-CTLs with 15 AA-long CDR3-TCRβ according to the CMV-reactivation pattern. **c** Difference in TCR-peptide binding scores among the major clones according to AA length of CDR3-TCRβ ($P = 0.003$ by the Kruskal–Wallis (K–W) test). **d** Difference in TCR-peptide binding scores among the major clones according to the presence of "(G)GG" motifs in CDR3-TCRβ ($P < 0.001$ by the Mann–Whitney U (M–W) test). **e** Difference in TCR-peptide binding scores among the major clones according to the CMV-reactivation pattern ($P = 0.31$ by the K–W test). **f** Difference in binding score weighted by the individual clone counts among the major clones in the early phase of allo-HCT according to the CMV-reactivation pattern ($P = 0.02$ by the K–W test, and $P = 0.0031$ by the J–T test for decreasing tendency). Individual box and whisker plots were constructed by the 25th percentile (Q1), median, and 75th percentile (Q3) with whiskers of 1.5 times interquartile range (IQR) lengths.

between the decreasing and increasing clones in the late phases. The binding scores of the increasing dominant clones in the late phases were significantly higher than those of the decreasing ones ($P = 0.016$, Fig. 3d).

The top 1 and 2 clones in the early phase of allo-HCT accounted for >75% in all of the no-CMV reactivation group, 78% of the one-episode group, and 43% of the repeated CMV reactivation group

($P = 0.018$ by Fisher's exact test). Those in the late phase accounted for >75% in 78% of the no-CMV reactivation group and 71% of the one-episode group, but in only 33% of the repeated CMV reactivation group, albeit this difference was not significant ($P = 0.21$).

Alternatively, if we focused on the clones that disappeared in the late phase, their proportion in the early phase increased in the

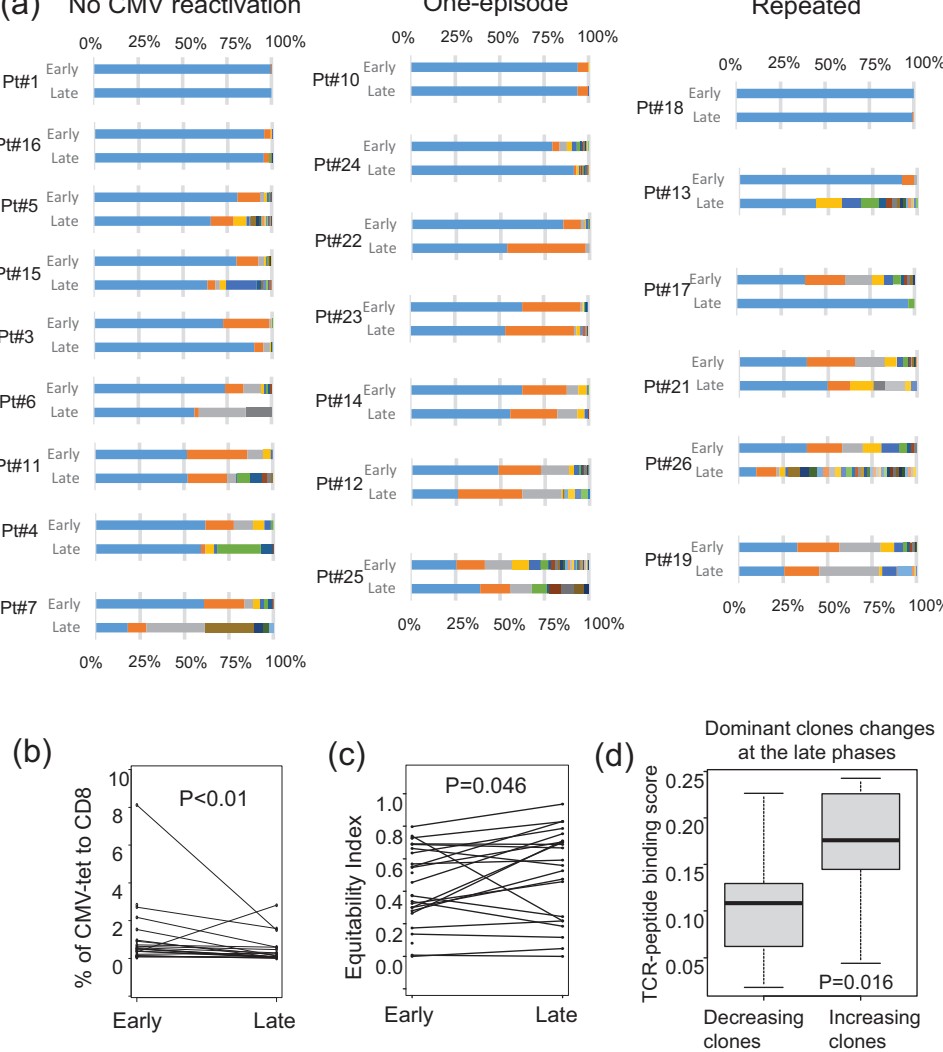

**Fig. 3 Changes in proportions and diversity of all HLA-A24-restricted CMV-pp65-specific cytotoxic T-cell (CMV-CTL) clones. a** Changes in the proportions of individual CMV-CTL clones between the early and late phases after allo-HCT in each patient for whom samples were available at both points (*n* = 22). The same colors suggest the same CMV-CTL clones between the phases within individual patients. **b** Changes in the proportions of CMV-CTLs in all groups pooled between the early and late phases of allo-HCT (*P* < 0.001 by Wilcoxon's signed-rank test). **c** Changes in Shannon's equitability index of CMV-CTLs in all groups pooled between the early and late phases of allo-HCT (*P* = 0.046 by Wilcoxon's signed-rank test). **d** Difference in TCR-binding scores of the dominant CMV-CTLs, which accounted for >35% of all CMV-CTLs within individual recipients in the early or late phases after allo-HCT, between the decreasing and increasing clones at the late phases (*P* = 0.016 by the Mann–Whitney test). Individual box and whisker plots were constructed by the 25th percentile (Q1), median, and 75th percentile (Q3) with whiskers of 1.5 times interquartile range (IQR) lengths.

following order: no-CMV reactivation < one-episode < repeated CMV reactivation groups (*P* = 0.026 by Jonckheere-Terpstra test for increasing tendency, Supplementary Fig. 4h). However, there was no statistical differences in increasing or decreasing tendency according to the CMV-reactivation pattern when we focused on the proportions of clones that newly appeared in the late phase (Supplementary Fig. 4i), although the proportions of newly appeared clones seemed larger in the repeated CMV reactivation group.

In summary, focusing on the dominant clones, the proportion of clones with higher TCR-binding scores tended to increase at the late phase, and conversely those with lower scores tended to decrease. Minor clones (as well as major clones in some cases) that had been detected in the early phase more frequently disappeared in the late phase in the repeated CMV reactivation group, while there was no significant difference in the proportion

of clones that newly appeared in the late phase according to the CMV-reactivation pattern.

**Homogeneity or heterogeneity according to individual HLA-A24-restricted CMVpp65-specific CTL clones within each recipient.** We next checked whether individual CMV-CTL clones had the same or different functions within each recipient (*n* = 3). The actually-analyzed cell counts with combined information on gene expression and TCRβ-CDR3 were 972 (Case 14, Fig. 4a), 747 (Case 4, Fig. 4b), and 501 (Case 3, Fig. 4c) cells after apparently dying cells were deleted. Based on their t-SNE clustering plots, the individual clones were homogeneously distributed (Fig. 4a–c), and there was no robustly significant difference with a two-fold change and *P* < 0.05 in GEP among the top 3 clones within each recipient, except for their TCR genes in all cases and *GNLY* and *FXYD5* in Case 3 (Supplementary Data 2–4).

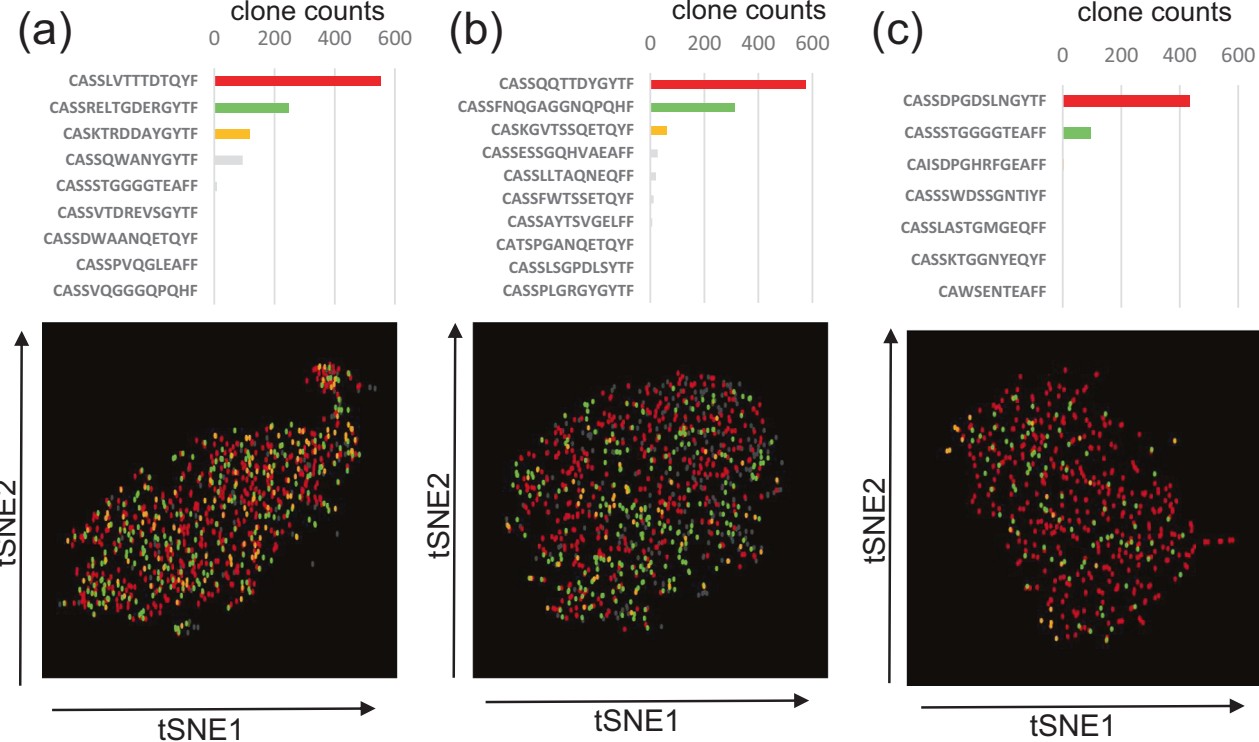

**Fig. 4 Single-cell RNA-sequencing of HLA-A24-restricted CMV-pp65-specific cytotoxic T-cells (CMV-CTL).** CMV-CTL clone counts and t-SNE plots in **a** Case14, **b** Case4, and **c** Case3. The actually-analyzed cell counts with combined information on gene expression and TCRβ-CDR3 were 972 (Case 14), 747 (Case 4), and 501 (Case 3) cells after apparently dying cells were deleted. Each dot represents a single cell and each color denotes the same individual clone. There was no robustly significant difference with a two-fold change and $P < 0.05$ in gene expression among the clones in each recipient, except for the TCR genes in all cases and *GNLY* and *FXYD5* in Case3. The t-SNE clustering plots are shown after deleting apparently dying cells from Case3.

Based on the relatively homogeneous distribution of GEP in the CMV-CTL clones within each recipient, we performed RNA-sequencing treating a bulk of CMV-CTLs, and compared GEP among recipients.

### GEP, protein–protein interaction (PPI) network, and gene ontology (GO) enrichment analyses of HLA-A24-restricted CMVpp65-specific CTLs in the early phase

*Comparison according to the CMV-reactivation pattern.* The explanation of individual genes described below was shown in Table 2 based on GeneCards (The Human Gene Database, https://www.genecards.org/). The top 200 differentially expressed genes (DEGs) with $P < 0.05$ are shown in Supplementary Data 5. They were well-clustered by the top 100 DEGs (Fig. 5a). Their PPI network was constructed using the top 100 DEGs (Fig. 5b). According to their degrees of centrality, meaning how many edges are connected to each gene node, *TNF* and *HELLS*[27] seemed to centrally work as hub genes in the network (Fig. 5b). In addition, *TNF* was further increased in both the no-CMV reactivation ($n = 8$) and one-episode ($n = 7$) groups compared with the repeated CMV reactivation group ($n = 6$) (Fig. 5b). The GO enrichment analysis demonstrated that the top 200 DEGs were involved in the "regulation of acute inflammatory response" and "regulation of viral genome replication" as an immunological response (Fig. 5c). In these processes, inflammatory genes such as *TNF* and *ISG20*[28] were shared among the biological terms. In addition, they were also involved in the processes of regulation of cell division, and other metabolic activities (Fig. 5c). In summary, *TNF* expression was increased in the no-CMV reactivation group, and cell replication /division would be more active, leading to the proliferation of CMV-CTLs in that group.

*Comparison according to donor CMV-serostatus.* The top 200 DEGs are shown in Supplementary Data 6. They were also well-clustered by the top 100 DEGs (Fig. 6a). The PPI network suggested that the increased genes of *CCR5*[29], *S100A8* and *S100A9*[30] centrally worked in the CMV-seropositive donor group ($n = 10$) (Fig. 6b). The top 200 DEGs were involved in "cellular defense response", "positive regulation of lymphocyte proliferation", and "production of molecular mediator involved in inflammatory response" as immunological processes (Fig. 6c). In these processes, inflammatory genes such as *CCR5*, *LGALS3*[31], and *IL17RA*[32] were identified as critical/shared genes among the biological terms. Inflammatory response and cell proliferation of CMV-CTLs might be more promptly increased in the CMV-seropositive donor group compared with the CMV-seronegative group.

*Comparison according to the CMV-reactivation pattern in the subgroup of CMV-seropositive or -negative donor.* Since we found that the GEP differed according to the donor CMV serostatus, we then checked the difference in GEP according to the CMV reactivation pattern in the sub-cohorts of donor CMV status. Focusing on the sub-cohort of recipients with CMV seropositive donors ($n = 10$), increased *TNF* still seemed to centrally work in the no-CMV reactivation group (Supplementary Fig. 5a–c, Supplementary Data 7). Focusing on the sub-cohort of recipients with CMV seronegative donors ($n = 11$), *CCR2*[29] and *IL7R* were increased in the repeated CMV reactivation group and reduced in the other groups (Supplementary Fig. 6, Supplementary Data 8).

Next, we focused only on GO terms of the immune system bioprocesses. In the CMV-seropositive donor cohort, GO enrichment analyses using the top 500 DEGs revealed that they were involved in "T-cell-mediated immunity" including cytokine

**Table 2 Gene names and their actual or hypothetical functions based on GeneCards (The Human Gene Database, https://www.genecards.org/).**

| Gene symbol | Gene name | Functional explanation |
|---|---|---|
| CCR2 | C-C Motif chemokine receptor 2 | a receptor for monocyte chemoattractant protein-1 |
| CCR5 | C-C Motif chemokine receptor 5 | a regulator of granulocytic lineage proliferation |
| CD160 | CD160 Molecule | the expression is closely associated with peripheral blood NK cells and CD8 T lymphocytes with cytolytic effector activity |
| CD80 | CD80 Molecule | a membrane receptor that is activated by the binding of CD28 or CTLA-4 |
| DDX5 | DEAD-Box helicase 5 | an RNA helicase |
| DLG5 | Discs large MAGUK scaffold protein 5 | a regulator of the Hippo signaling pathway involved in regulating cell proliferation |
| EPHB6 | EPH Receptor B6 | a kinase-defective receptor for members of the ephrin-B family that inhibits JNK activation and TCR-induced IL-2 secretion |
| FCGR2A | Fc Fragment Of IgG receptor IIa | a low affinity receptor for immunoglobulin gamma, promoting phagocytosis and cellular responses against pathogens |
| HBB | Hemoglobin subunit beta | an oxygen transporter |
| HELLS | Helicase, lymphoid specific | a lymphoid specific helicase and regulator for the expansion or survival of lymphoid cells |
| IL17RA | Interleukin 17 receptor A | a receptor of IL17 (a proinflammatory cytokine) secreted by activated T-lymphocytes |
| IL18 | Interleukin 18 | a proinflammatory cytokine primarily involved in polarized T-helper 1 cell and NK cell immune responses |
| IL1R1 | Interleukin 1 receptor Type 1 | a receptor of an important mediator, IL1, involved in many cytokine-induced immune and inflammatory responses |
| IL6R | Interleukin 6 receptor | a subunit of the interleukin 6 (IL6) receptor complex) |
| IL7R | Interleukin 7 receptor | a receptor for IL7 |
| ISG20 | Interferon stimulated exonuclease gene 20 | an interferon-induced antiviral exoribonuclease |
| LAX1 | Lymphocyte transmembrane adaptor 1 | a negative regulator of TCR-mediated signaling in T-cells and BCR (B-cell antigen receptor)-mediated signaling in B-cells |
| LGALS3 | Galectin 3 | a pre-mRNA splicing factor in acute inflammatory responses |
| LILRB1 | Leukocyte immunoglobulin like receptor B1 | a member of the leukocyte immunoglobulin-like receptor family, transducing inhibitory signals and down-regulation of the immune response |
| LILRB2 | Leukocyte immunoglobulin like receptor B2 | a member of the leukocyte immunoglobulin-like receptor family, transducing a negative signals and inhibiting stimulation of an immune response |
| LY96 | Lymphocyte antigen 96 | a protein that is associated with toll-like receptor 4 on the cell surface and confers responsiveness to lipopolysaccharide |
| MAD2L1 | Mitotic arrest deficient 2 Like 1 | a component of the mitotic spindle assembly checkpoint) |
| RAD51AP1 | RAD51 Associated protein 1 | a structure-specific DNA-binding protein involved in DNA repair |
| RSAD2 | Radical S-adenosyl methionine domain containing 2 | an interferon-inducible antiviral protein that belongs to the S-adenosyl-L-methionine (SAM) superfamily of enzymes |
| S100A8 | S100 Calcium binding protein A8 | a regulator of inflammatory processes and immune response |
| S100A9 | S100 Calcium binding protein A9 | a regulator of inflammatory processes and immune response |
| SNAP25 | Synaptosome associated protein 25 | a regulator of neurotransmitter release |
| TNF | Tumor necrosis factor | a multifunctional proinflammatory cytokine |
| TNFRSF21 | TNF Receptor superfamily member 21 | a negative regulator of T-cell responses triggered by TCR stimulation |
| TRAC | T Cell receptor alpha constant | a constant region of TCRα chain |
| TUBB | Tubulin beta class I | a beta tubulin protein |
| UBE2C | Ubiquitin conjugating enzyme E2 C | a member of the E2 ubiquitin-conjugating enzyme family |
| VEGFA | Vascular endothelial growth factor A | a growth factor active in angiogenesis, vasculogenesis and endothelial cell growth |

production, "T-cell costimulation", "negative regulation of lymphocyte activation", and "mast cell-mediated immunity" (Fig. 7a). The shared genes associated with cytokine production such as *TNF*, *IL1R1*[33], and *RSAD2*[34] were increased in the no-CMV reactivation group (Fig. 7b). In addition, the genes associated with negative regulation of T-cell activation such as *LAX1*[35], *TNFRSF21*[36], and *DLG5*[37] were also increased in the no-CMV reactivation group (Fig. 7b). On the other hand, *LILRB2*[38] was increased in the repeated CMV reactivation group (Fig. 7b).

In the CMV-seronegative donor cohort, the GO enrichment analyses using the top 324 DEGs with a false discovery rate (FDR) < 0.4 revealed that they were involved in immunological bioprocesses of "lymphocyte proliferation", and "regulation of T-cell activation" (Fig. 7c). The lymphocyte proliferation process included not only the regulation of T-cell but also the regulation of B-cell and macrophage chemotaxis. The genes associated with T-cell costimulation such as *CD160*[39] and *CD80*[40] were increased in the no-CMV reactivation group (Fig. 7d). On the other hand,

*LILRB1*[41], as well as *CCR2* and *IL7R*, was increased in the repeated CMV reactivation group (Fig. 7d).

In summary, when we focused on the CMV-seropositive donor cohort, inflammatory cytokine production such as *TNF* would be increased, but further or excess T-cell activation would be suppressed by the genes like *LAX1* in the no-CMV reactivation group. On the other hand, when we focused on the CMV-seronegative donor cohort, T-cell costimulation signaling of *CD160* and *CD80* would be increased in the no-CMV reactivation group, while a relative lack of efficiently-activated CMV-CTLs may be suggested by *CCR2* and *IL7R* in the repeated CMV reactivation group.

**GEP, PPI, and GO enrichment analyses of HLA-A24-restricted CMVpp65-specific CTLs in the late phases after allo-HCT**
*Comparison according to the CMV-reactivation pattern.* The top 200 DEGs are shown in Supplementary Data 9. In the clustering heatmap by the top100 DEGs, they did not seem to be clearly

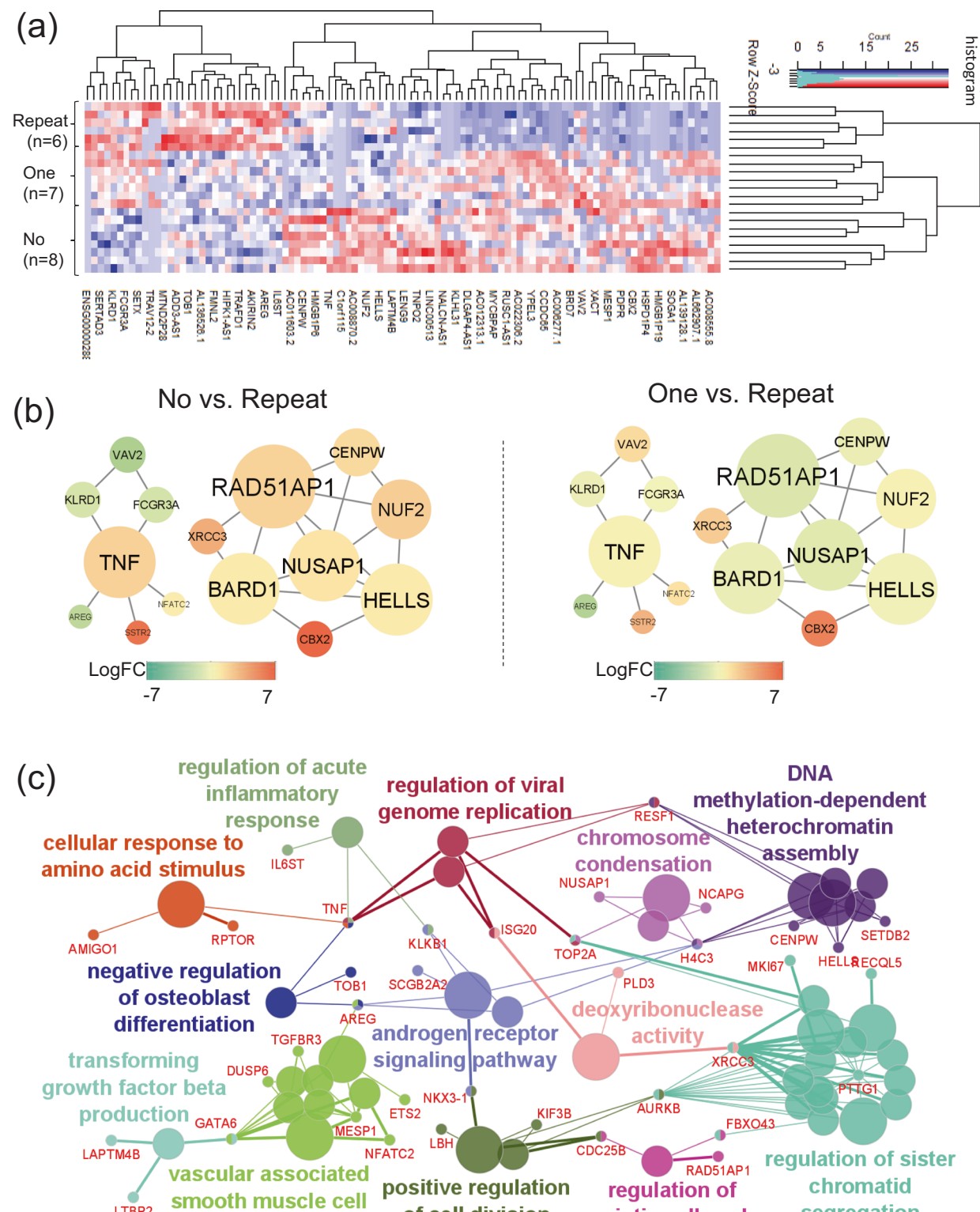

**Fig. 5 Gene expression profile (GEP), protein–protein interaction (PPI) network, and gene ontology (GO) enrichment analyses in HLA-A24-restricted CMV-pp65-specific cytotoxic T-cells (CMV-CTL) according to the CMV reactivation pattern in the early phase of allo-HCT. a** A clustering heatmap of GEP using the top 100 differentially expressed genes (DEGs) according to the CMV-reactivation pattern: no-CMV reactivation (no-group, $n = 8$), one-episode of CMV reactivation (one-group, $n = 7$), and repeated episodes of CMV reactivation (repeated-group, $n = 6$). **b** PPI network constructed by the top 100 DEGs which had close connections with each other. The shape size suggests the degree of centrality of the PPI network, meaning how many edges are connected to each gene node. The heat color denotes the log-fold changes in individual gene expression between the no- vs. repeated- groups or between the one- vs. repeated groups. Only the networks with ≥5 connections are shown. **c** GO and the shared genes derived from the top 200 DEGs. Each circle denotes an identified term with a $P$-value of <0.05 without the Bonferroni correction. The same colors mean GO terms that belong to the same GO term-tree groups. Only the names of the leading GO terms with highest significance in each GO term-tree group are shown.

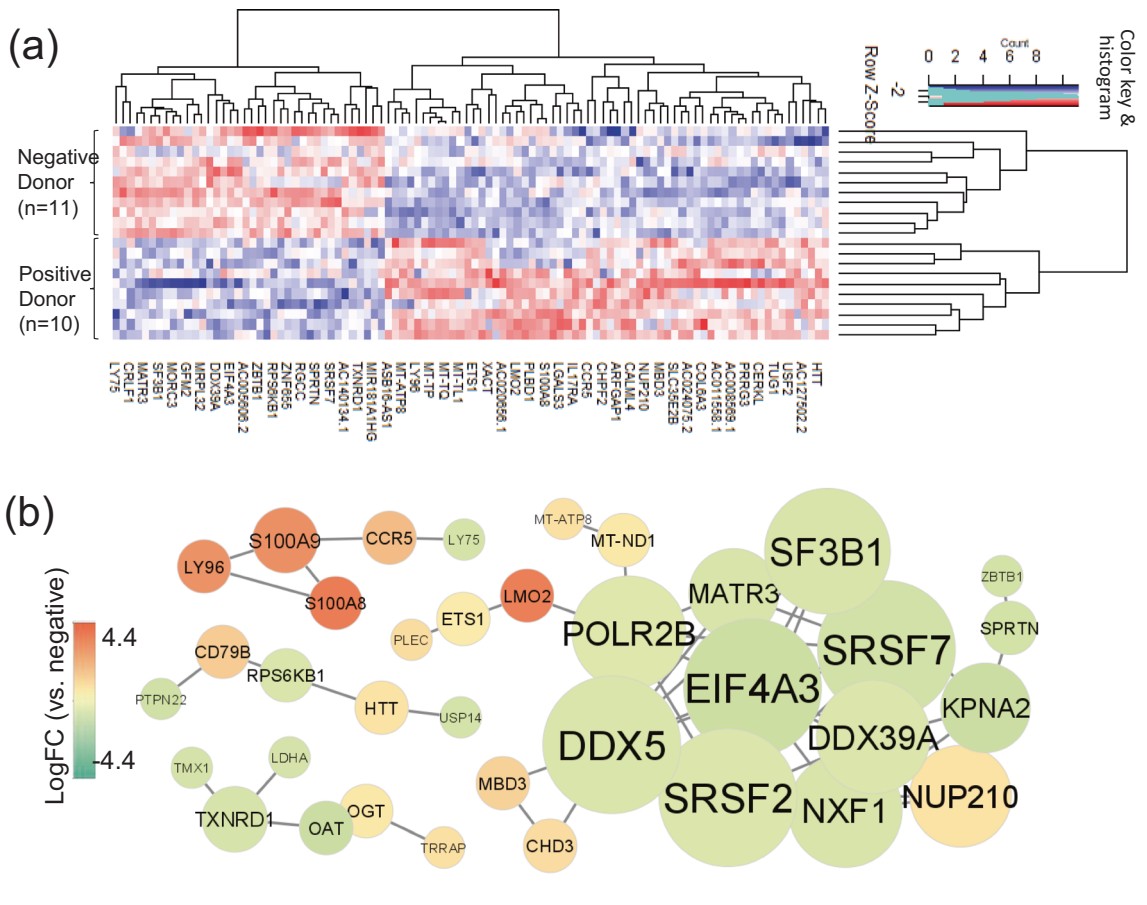

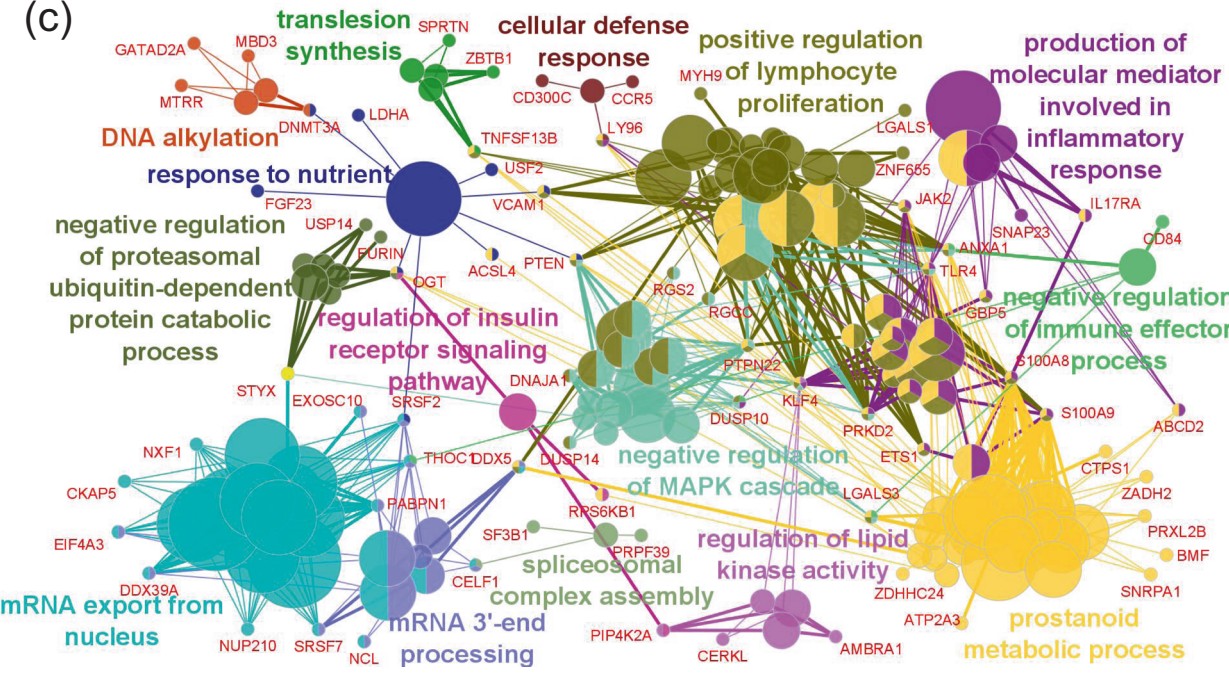

divided according to the CMV reactivation pattern compared with that in the early phase (Fig. 8a). We checked their CMV reactivation, disease status, and GVHD at the sampling of the late phase. The split subgroup of the no-CMV reactivation group ($n = 2$) experienced subsequent hematological or molecular relapse. In addition, 2 of 3 recipients in the split subgroup of the repeated-CMV reactivation group still suffered from CMV

reactivation and severe GVHD. However, the DEGs were not clearly divided when we considered these clinical backgrounds. The PPI network demonstrated that immune response-related genes like *FCGR2A*[42] and cell replication-related genes like *MAD2L1*[43] seemed to work as hub genes in the network (Fig. 8b). Focusing on the GO terms of immune systems, the top 200 DEGs were involved in the processes of αβ T-cell activation, NK cell

**Fig. 6 Gene expression profile (GEP), protein–protein interaction (PPI) network, and gene ontology (GO) enrichment analyses in HLA-A24-restricted CMV-pp65-specific cytotoxic T-cells (CMV-CTL) in the early phase of allo-HCT according to the donor CMV serostatus. a** A clustering heatmap of GEP using the top 100 differentially expressed genes (DEGs) between CMV-seronegative ($n = 11$) and -seropositive ($n = 10$) donors. **b** PPI network constructed from the top 100 DEGs which had close connections with each other. The shape size suggests the degree of centrality of the PPI network, meaning how many edges are connected to each gene node. The color heat denotes log-fold changes in individual gene expression between the CMV-seropositive vs. -seronegative donor groups. Only networks with ≥5 connections are shown. **c** GO and the shared genes derived from the top 200 DEGs. Each circle denotes an identified term with a *P*-value of <0.05 without the Bonferroni correction. The same colors mean GO terms that belong to the same GO term-tree groups. Only the names of the leading GO terms with the highest significance in each GO term-tree group are shown.

immunity, and leukocyte cytotoxicity (Fig. 8c). The shared genes such as *TRAC*, *IL18*[44], and *IL6R*[45] for the immunological processes remained increased in the repeated CMV reactivation group (Fig. 8d), suggesting that CMV-CTLs might still have to work actively in the repeated CMV reactivation group even in the late phase after allo-HCT and to recruit other immune cells, while those in the other two groups would be under a steady state without CMV reactivation for a long time. Alternatively, the CMV-CTLs themselves in the repeated CMV reactivation group may be ineffective against the host CMV but be capable of signaling NK cells, B cells and components of the innate immune system.

*Comparison according to donor CMV-serostatus.* The top 200 DEGs are shown in Supplementary Data 10. There was only one gene with FDR < 0.05 and three genes with FDR < 0.4, suggesting that there was no robust difference in GEP in the comparison in the late phase after allo-HCT. Therefore, PPI and GO enrichment analyses were not performed.

## Discussion

Here we demonstrated the features of diversity and binding scores in TCRβ-CDR3 and the heterogeneity in gene expression of HLA-A24-restricted CMVpp65-specific CTLs following allo-HCT.

A skewed preference of BV or BJ genes was observed as well as preferential AA motifs in TCRβ-CDR3 in HLA-A24-restricted CMV-CTLs. The preferential selection of BV and BJ genes has been reported in various CTLs, and it differs according to the target antigens and HLA[46–50]. For example, HLA-A02-restricted CMVpp65-specific CTLs were reported to preferentially select BV7 and BV12 with BJ1-4 and BJ1-2[46]. HLA-A24-restricted HTLV-1 tax-specific CTLs exclusively selected BV7[47,49], and HLA-A02-restricted influenza A (IFA)-specific CTLs distinctly used BV19[46]. A preferential selection bias was actually found in this study. Overall, less-restricted BV gene usage (20–30%) was observed in CMV-CTLs regardless of HLA, compared with that in HTLV-1 or IFA-specific CTLs (50–85%)[46,47,49]. This may suggest that CMV-CTLs are more functionally diverse than other virus-specific CTLs[46]. In addition, the AA-length preference of TCRβ-CDR3 in HLA-A2 CTLs against CMV (most frequent AA length = 14) has been reported to be quite different from that against IFA (most frequent length = 11)[46]. Actually, the preferred AA length of HLA-A24 CMV-CTLs was around 15. Especially, clones with a longer AA length of TCRβ-CDR3 were more prevalent among the major clones in the no-CMV reactivation group than among those in the repeated CMV reactivation group. Furthermore, preferred AA motifs also seemed to differ according to AA length and the CMV-reactivation pattern. Especially, the "GGG" motif in TCRβ-CDR3 with 15 AA seemed to be frequently selected in the no-CMV reactivation group. However, HLA-A24 CMV-CTLs did not show a common AA motif that was observed in every recipient, such as the "PDR" motif in HLA-A24 HTLV-1-CTLs[47,49].

Next, we found increased TCR-peptide binding scores in TCRβ-CDR3 with AA lengths of 15 and 17, as well as in that with a "(G)GG" motif. In general, CTLs with high affinity toward antigens demonstrate an increased cytotoxic function[48,49,51,52]. CMV-CTLs with a higher binding score could be candidates for adoptive T-cell therapies for refractory CMV diseases, although the actual binding affinity of whole TCR-peptide-MHC complex and its specific cytotoxicity should be confirmed after individual clones are established. Alternatively, the dominant CMV-CTL clones in the no-CMV reactivation group even under steroid treatment for aGVHD (e.g., Case3 in Supplementary Data 1) are another candidate for adoptive T-cell therapies, since aGVHD and steroid therapy has been established as a risk factor for CMV reactivation[53]. Notably, the binding scores weighted by the cell counts of individual clones were increased in the no-reactivation group (Fig. 2f), although clones with a higher binding score equally appeared in the repeated CMV reactivation group (Fig. 2e). Thus, CMV reactivation post allo-HCT would be influenced not only by the TCR sequence itself, but also by whether T-cells could properly proliferate and functionally work under immunosuppressive conditions post allo-HCT. Actually, the RNA-sequence of CMV-CTLs suggested that their DEGs according to the CMV reactivation pattern and donor CMV serostatus were involved in the network associated with the effector cytokines like *TNF* or in that of cell division. Therefore, in addition to the TCR sequence/structure analyses, further investigations of efficient cell proliferation, differentiation, cytokine production, and senescence using actual recipient samples will be required for the optimized control of CMV reactivation by CMV-CTLs.

In this study, more minor clones tended to appear and disappear in the repeated CMV reactivation group than in the other groups. Consistent with this finding, a Swiss group recently reported that entire T-cell reconstitution and diversity after allo-HCT were strongly affected by the CMV serostatus of the donor and recipient as well as CMV reactivation, and the cumulative frequencies of CMV-specific clones were increased in recipients with CMV reactivation[13]. In addition, the frequency of CMV-CTLs significantly decreased, while the diversity increased in the late phase of allo-HCT, compatible with a previous report[51].

We specifically demonstrated the difference in GEP of CMV-CTLs among recipients according to the CMV-reactivation pattern and donor CMV serostatus. As expected, inflammatory cytokines like *TNF* were increased in CMV-CTLs of the no-CMV reactivation group in the early phase of allo-HCT. According to the clone dominance observed between a donor and recipient post allo-HCT in the no-CMV reactivation group with CMV-positive donors as mentioned above (Supplementary Fig. 4f), the immunity transferred from CMV seropositive donors was considered to mainly keep playing a role against CMV throughout allo-HCT in this group. On the other hand, the immunity against CMV should be naively introduced after allo-HCT from CMV-seronegative donors. The CMV-CTLs transferred from CMV-seropositive donors are generally considered to be "older", while CMV-CTLs naively introduced from CMV-negative donors after

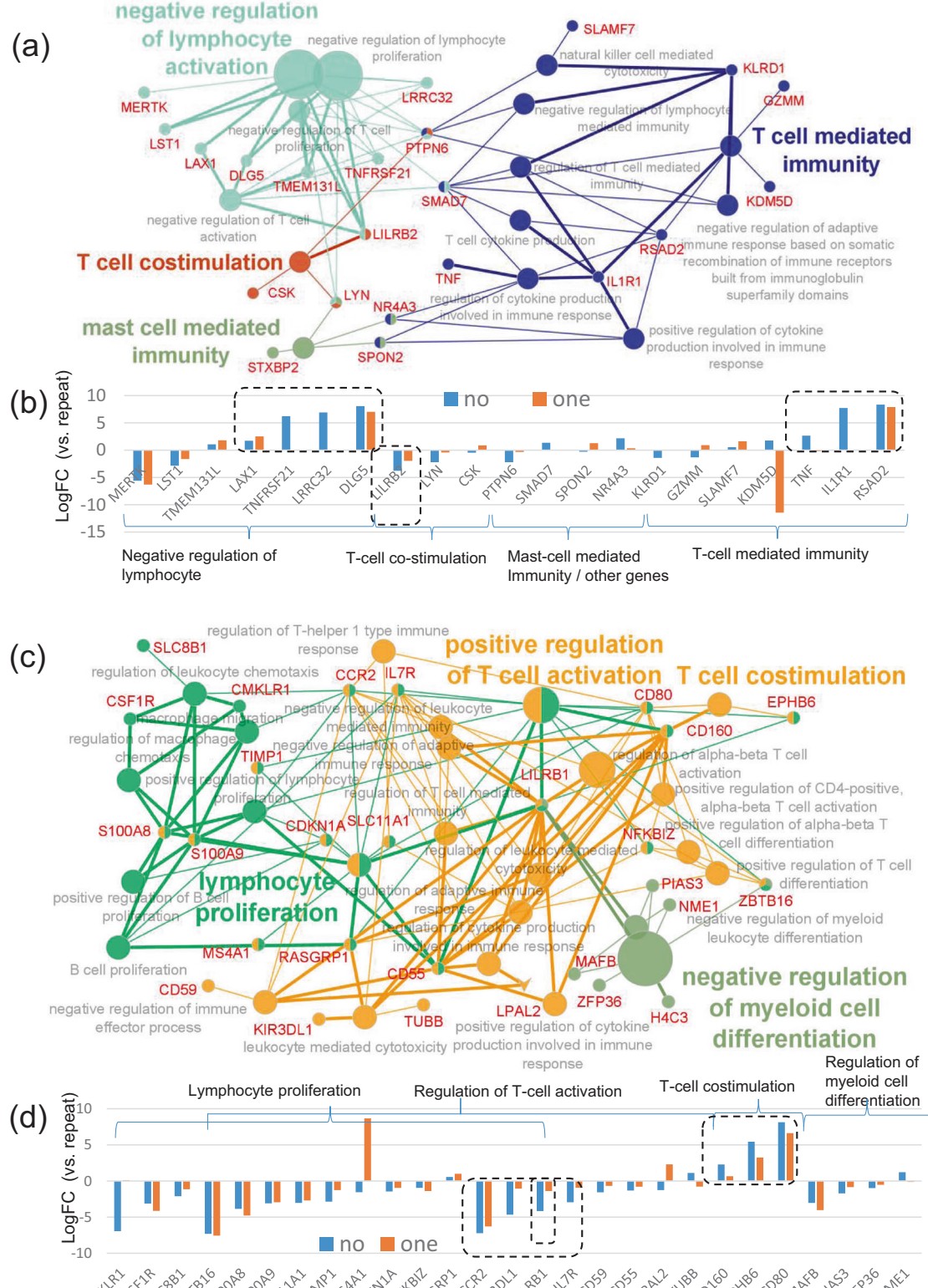

allo-HCT should be "younger". Thus, we initially expected that the CMV-CTLs in the CMV-seropositive donor group might have a lower expression of *CD28* since reduction or loss of *CD28* is known to be an indicator of cell senescence or exhaustion[54,55]. However, we failed to detect the difference in *CD28* expression between the CMV-seropositive and -negative donor groups. Therefore, CMV-CTLs in the CMV-seropositive donor group

may include functional stem-memory CMV-CTLs for efficient self-renewal and smooth transition to effector CTLs as reservoirs of highly functional memory T-cells[52,56], and prevent exhaustion of overall CMV-CTLs. If we consider the DEGs according to the CMV reactivation pattern only in the CMV-seropositive donor cohort, the negative regulation of T-cell activation by several genes like *LAX1*[35] and *TNFRSF21*[36] as well as cytokine

**Fig. 7 Gene ontology (GO) enrichment analyses focusing on the immune system processes in HLA-A24-restricted CMV-pp65-specific cytotoxic T-cells (CMV-CTL) in the early phase of allo-HCT according to the CMV reactivation pattern in the subgroups of donor CMV serostatus. a** GO of the immune system processes and the shared genes derived from the top 500 differentially expressed genes (DEGs) with FDR < 0.1, focusing on the CMV-seropositive cohort alone. Each circle denotes an identified term with a *P*-value of <0.05 without the Bonferroni correction. The fused GO terms are shown. The same colors mean GO terms that belong to the same GO term-tree groups. The names of the leading GO terms are shown, based on the number of genes in each GO term-tree group. **b** The log-fold changes in the shared genes in the no-CMV (no-group, *n* = 5) and one-episode (one-group, *n* = 3) reactivation groups compared with the repeated CMV reactivation group (repeated group, *n* = 2). **c** GO of the immune system processes and the shared genes derived from the top 324 DEGs with FDR < 0.4, focusing on the CMV-seronegative cohort alone. Each circle denotes an identified term with a *P*-value of <0.05 without the Bonferroni correction. The fused GO terms are shown. The same colors mean GO terms that belong to the same GO term-tree groups. The names of the leading GO terms are shown based on the number of genes in each GO term-tree group in addition to the term of "T-cell costimulation". **d** The log-fold changes in the shared genes in the no-CMV (no-group, *n* = 3) and one-episode (one-group, *n* = 4) reactivation groups compared with the repeated CMV reactivation group (repeated group, *n* = 4).

production pathways were enriched in the no-CMV reactivation group. The negative regulation of T-cell activation might be simultaneously upregulated to prevent the excessive exhaustion of CMV-CTLs. When we focused on the CMV-seronegative donor cohort, *CD160* and *CD80* were increased in the no-CMV reactivation group. There is some debate regarding whether *CD160* on T-cells have costimulatory or coinhibitory signals. The expression of *CD160* on NK-cells has been reported to trigger cytotoxic activity[39], and that on CD8+ T-cells is considered to costimulate the proliferation of activated T cells[57], indicating the costimulatory potential of *CD160* signaling. On the other hand, in the setting of chronic viral infection such as HIV, *CD160* signaling was reported to induce exhaustion and functional impairment specific to influenza, EBV, and CMV[58]. Recently, in pancreatic cancer, *CD160* expression on CD8+ T-cells is reported to have active effector responses but limited activation potential[59]. Taken together, *CD160* might help to control the acute phase of CMV reactivation, but would not be beneficial if the viral stimulation persisted for a long time. *CD80* is mainly expressed on antigen-presenting cells, but is also detected on CD8+ T-cells. The activation of *CD80* by the binding of *CD28* or *CTLA-4* induces both stimulatory and inhibitory signaling. Recently, *CD80* expression on memory CD8+ T-cells after acute viral infections has been reported to play an important role in suppressing excessive CD8+ T-cell recall responses, leading to an appropriate recall immune response[40]. The expression of *CD80* might also be favorable for the no-CMV reactivation group. On the other hand, the expression levels of *IL7R*, *CCR2*, and *LILRB1* were increased in the repeated CMV reactivation group of the CMV-seronegative donor cohort. Since *IL7R* expression is known to be reduced on activated virus-specific effector T-cells[54,60] and *CCR2* is reported to be downregulated in memory /effector T-cells following TCR stimulation[29], the increased *IL7R* and *CCR2* expression in the repeated CMV reactivation group may suggest a relative lack of efficiently-activated and -proliferated CMV-CTLs compared to those in the other two groups. Interestingly, inhibitory receptors like *LILRB1* (also known as *CD85j*) and *LIRRB2* (also known as *CD85d*) were increased in the repeated CMV reactivation group in the CMV-seronegative and -positive donor cohorts, respectively. *LILRB1* has been reported to be mainly expressed on the terminally-differentiated effector T-cells and to increase with age[41], suggesting a kind of cell senescence. Therefore, repeated CMV reactivation after allo-HCT may promote cell senescence of CMV-CTLs. The inhibition of *LILRB1* has been reported to promote /enhance the efficient proliferation of CMV-CTLs[41]. In addition, both of *LILRB1* and *LILRB2* have recently been considered as innate and adaptive immune checkpoint molecules[61]. Targeting these increased receptors might be a candidate of future treatment strategies to recruit of CMV-CTLs for the repeated CMV reactivation group. It would be warranted to investigate the association of these identified DEGs with efficient CTL function

using actual patient samples in the context of allo-HCT and CMV reactivation.

This study had several limitations. First, it might be too small to draw any definitive conclusions regarding the association of the repertoire diversity with clinical outcomes. Second, we focused on only HLA-A24-restricted CMV-CTLs. Other class-I HLA-restricted CMV-CTLs could also play a role against CMV, which might have contributed to the lower proportion of tetramer-positive cells in patients with recurrent CMV viremia. We must assess the overall repertoire diversity beyond HLA restriction to clarify clinical aspects in the future. Third, we analyzed only TCRβ-CDR3, and not TCRα. Therefore, the actual binding affinity might be different from the predicted score in each T-cell. Our study was based on observational facts. Functional analyses would be warranted in the future, if all CMV-CTL clones could be established and all pairs of TCRα and TCRβ could be linked in individual clones. Our future studies need to explore the logical hypotheses whether the maintenance of the dominant clones in the no-CMV reactivation group can also actually control CMV reactivation in the repeated group, and whether we can predict a high-risk patient for multiple CMV reactivations by analyzing pre-transplant donor cells. However, this study primarily sought to provide an in-depth profiling of clinically transferred T-cells, and demonstrated the features of GEP in CMV-CTLs following allo-HCT according to the CMV-reactivation pattern as well as those of AA motifs used in HLA-A24 CMV-CTLs. Thus, these results could contribute to the expansion of public databases, leading to future investigations.

In conclusion, the current study shed light on the features of CMV-CTLs in terms of not only repertoire diversity but also TCR-peptide binding potential and GEP according to the CMV-reactivation pattern after allo-HCT. These methods and results can help us to better understand immune reconstitution following allo-HCT, leading to future studies on the clinical application of adoptive T-cell therapies.

## Methods

**Study design**. The research objective was to clarify the features of the AA component of TCRβ-CDR3, and to reveal TCR diversity and GEP differences in HLA-A24 CMV-CTLs according to the CMV reactivation pattern among recipients after allo-HCT. The current laboratory experiments included TCR determination of CMV-CTLs by NGS for the assessment of diversity, single-cell RNA-sequencing of isolated CMV-CTLs for the assessment of homogeneity within each case, and bulk RNA-sequencing of CMV-CTLs for comparison according to the CMV-reactivation pattern. Several bioinformatics tools were applied to assess TCR features, estimate peptide-binding scores of TCRβ-CDR3. This study included CMV-seropositive recipients who received allo-HCT in our institution between 2008 and 2016, and who survived for ≥6 months without disease relapse. All of the recipients and their corresponding donors had to have HLA-A24:02 or -A24:20. In addition, we analyzed three related donors whose samples before allo-HCT were available (*n* = 3). Peripheral blood samples were obtained from recipients every 1–3 months after allo-HCT, whereas those from the donors were obtained during mobilization by granulocyte-colony stimulating factor. Patient blood samples before hematological relapse were required for this study. Eventually, we analyzed TCR of

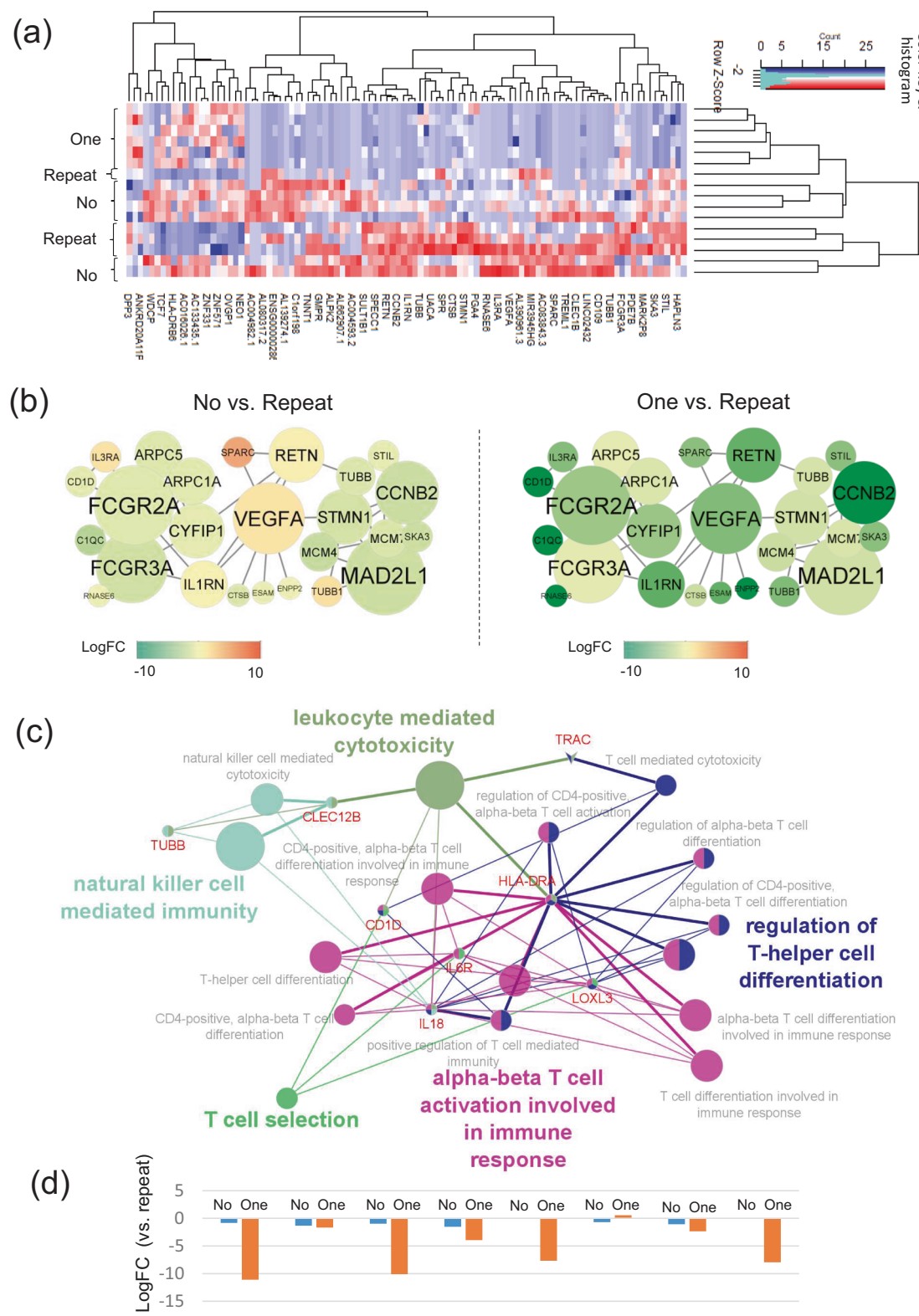

CMV-CTLs in 51 samples obtained from 26 patients and 3 donors (Table 1), including 2 samples analyzed by the direct single-cell method alone and reported previously[9,10]: 26 samples in the early phase (*n* = 7 at 1 month, 3 at 2 months, and 16 at 3 months after allo-HCT), and 22 samples in the late phase (*n* = 2 at 6 months, 19 at 1 year, and 1 at 2 years after allo-HCT). This study was approved by the institutional review board of Jichi Medical University and all subjects gave their written informed consent for the cryopreservation and analysis of blood samples in accordance with the Helsinki declaration.

**Category definition and preemptive treatment for CMV reactivation.** CMV reactivation was monitored weekly after neutrophil engraftment using a CMV antigenemia assay by the C10/11 method[7,62]. Reactivation was defined when ≥3 CMV antigenemia were detected. CMV reactivation was categorized into three patterns: no reactivation, one sequential reactivation event, and repeated episodes even after CMV reactivation was once resolved by ganciclovir treatment. The recipients were categorized according to donor CMV serostatus (positive vs. negative) and/or the CMV-reactivation pattern (no vs. one vs. repeated). As

**Fig. 8 Gene expression profile (GEP), protein–protein interaction (PPI) network, and gene ontology (GO) enrichment analyses in HLA-A24-restricted CMV-pp65-specific cytotoxic T-cells (CMV-CTL) according to the CMV reactivation pattern in the late phase of allo-HCT. a** A clustering heatmap of GEP using the top 100 differentially expressed genes (DEGs) according to the CMV-reactivation pattern: no-CMV reactivation (no-group, $n = 6$), one-episode of CMV reactivation (one-group, $n = 6$), and repeated episodes of CMV reactivation (repeated-group, $n = 4$). **b** PPI network constructed by the top 100 DEGs which had close connections with each other. The shape size suggests the degree of centrality of the PPI network, meaning how many edges are connected to each gene node. The color heat denotes the log-fold changes in individual gene expression between the no- vs. repeated groups or between the one- vs. repeated groups. Only networks with ≥5 connections are shown. **c** GO and the shared genes derived from the top 200 DEGs. Each circle denotes an identified term with a *P*-value of <0.05 without the Bonferroni correction. The same colors mean GO terms that belong to the same GO term-tree groups. Only the names of the leading GO terms with the highest significance in each GO term-tree group are shown. **d** The log-fold changes in the shared genes in the no- and one-groups compared with the repeated group.

---

preemptive therapy, ganciclovir was administered in an initial dose of 5 mg/kg/day, and the dose was sequentially adjusted according to renal function and changes in CMV antigenemia [7,8,63].

**Staining and sorting for HLA-A24-restricted CMV-pp65 CTLs.** Mononuclear cells in peripheral blood (PBMCs) were isolated by density gradient sedimentation using Lymphoprep (Axis-Shield PoC AS) and cryopreserved at −80 °C or in liquid nitrogen until use. PBMCs were incubated with HLA-A*2402 CMV-pp65$_{(341-349)}$ peptide (QYDPVAALF)-binding HLA tetramer-PE (CMV-tetramer) (10 μL, TS-0020-1C, Medical & Biological Laboratories) for 30 min at room temperature, and then stained with anti-human CD3-FITC (5 μL, 317306, OKT3, BIoLegend) or CD3–APC(5 μL, 317318, OKT3, Biolegend), CD8a-APC (5 μL, 301014, RPA-T8, BIoLegend) or CD8a-FITC (5 μL, 301006, RPA-T8, BIoLegend), CD45RA-PECy7 (5 μL, 304128, HI100, BIoLegend), CCR7-APCCy7 (5 μL, 353226, G043H7, BIo-Legend) and 7AAD (420404, BIoLegend). HLA-A*2402-restricted CMV-pp65 CTLs (CMV-CTL) were defined as CD3+CD8+CMV-teramer+ T-cells. Stained PBMCs were subjected to flow cytometry analyses, and CMV-CTLs were sorted with FACSAria™ II (BD Biosciences) and analyzed by Diva software (BD Biosciences) (Supplementary Fig. 1a).

**Smarter-NGS method for TCR determination.** Up to 1,000 CMV-CTLs were sorted directly into a PCR tube per sample with lysis buffer (Supplementary Table 1). After sorting, CMV-CTLs were treated by a SMARTer® Human TCR a/b Profiling Kit (Takara Bio, Inc.) according to the manufacturer's instructions with modified thermal cycles using Veriti (Thermo Fisher Scientific). Briefly, first-strand cDNA was synthesized after direct sorting and cell lysis. TCRβ-CDR3 in each cell was amplified by 2 steps of polymerase chain reaction (PCR) with 30 cycles for PCR1 and 20 cycles for PCR2, respectively. Sequencing libraries were generated following solid-phase reversible immobilization bead purification (SPRI purification) using an Agencourt AMPure XP PCR purification kit (Beckman Coulter). Thereafter, electrophoresis analyses were performed to validate whether library production, purification, and two steps of size selection succeeded, using an Agilent 2100 Bioanalyzer (Agilent Technologies) with Agilent DNA1000 reagent (Agilent Technologies). If a peak was detected in the range around 700–800 bp (Supplementary Fig. 7a), the synthesized cDNA libraries were transferred to Takara Bio, Inc. (Otsu, Japan) for sequencing with a Miseq Reagent Kit v3 (Illumina) and a PhiX control Kit v3 (Illumina) on an Illumina MiSeq sequencer (Illumina) with paired-end, $2 \times 300$ base pair reads. After the reads were analyzed with MiSeq Control Software (MCS) v2.6.2.1, Real Time Analysis (RTA) v1.18.54, and bcl2fasq2 v2.17, the results were re-transferred to Jichi Medical University for further analyses. In this study, the individual clones were considered to be an error and ignored if clone proportion (%) × sorted cell counts <0.5 cells.

**Direct single-cell method for TCR determination.** We compared the proportion of CMV-CTL clones identified by the current Smarter-NGS method with the TCR data determined by the direct single-cell method reported in our previous studies[9,10]. In that method (Supplementary Table 2), individual CMV-CTLs were sorted into PCR tubes at a single-cell level, one cell per tube. After direct cell lyses, cDNA of TCRβ-CDR3 was synthesized by reverse transcription (RT), and two sequential steps of semi-nested PCR were performed to identify the BV family of individual cells using 24 kinds of TCR-β variable region (BV) gene family-specific primers[9,10,49,64] and 2 kinds of BC primers[9,10,49,64]. Thereafter, we identified the specific BV genes of individual cells, and directly sequenced AA of V-D-J CDR3 of the individual T-cells (Supplementary Table 2)[9,10,49,64]. Donor clones of Case3 and late-phase clones of Case 26 had been analyzed only by the direct-single cell analysis method and reported in our previous paper [9].

**Circos plot and sequence logo for CDR3 in HLA-A24-restricted CMV-pp65 CTLs.** A circos plot was made to visualize the relationship between BV and BJ genes in CDR3 of CMV-CTLs through an online service (http://mkweb.bcgsc.ca/tableviewer/). AA sequence logos were generated to graphically represent AA of

CDR3 in CMV-CTLs through the WebLogo online service (https://weblogo.berkeley.edu/logo.cgi) [65].

**Statistics and reproducibility.** Shannon's equitability (or Pielou's evenness index) $(E_H)$ was calculated to assess the diversity of CMV-CTLs in individual samples as follows, using the "vegan" package (ver. 2.5.6):

$$H = -\sum_{i=1} P_i Log(P_i), P_i = proportional\ abundance\ of\ clonotypes\ i \quad (1)$$

$$E_H (or\ J') = H/log S, S = total\ number\ of\ clonotypes\ in\ the\ group \quad (2)$$

When only one clone was observed, $E_H$ was set to zero in this analysis.

The Pearson's correlation test was used to check the association between two numerical variables. The Jonckheere–Terpstra test was used to assess the increasing or decreasing tendency of numerical variables according to the reactivation pattern, while the Mann–Whitney or Kruskal–Wallis test was used for simple comparisons of numerical variables. The Wilcoxon's signed-rank test was used to identify time-dependent changes in numerical variables between the early and late phases after allo-HCT. When a *P*-value of less than 0.05 was obtained, we considered that there was statistical significance. All clinical and statistical approaches were conducted using EZR ver1.54 (Jichi Medical University at http://www.jichi.ac.jp/saitama-sct/SaitamaHP.files/statmedEN.html) [66].

**Estimation of TCR-peptide binding score of TCR-CDR3.** The binding affinity between individual TCRβ-CDR3 and CMV-pp65 peptide (QYDPVAALF) was predicted using the ERGO system (pEptide tcR matchinG predictiOn) (http://tcr.cs.biu.ac.il/, accessed in July, 2020)[26]. The TCR-peptide binding scores were obtained by the TCR autoencoder-based model trained by VDJdb[67,68], an open, comprehensive database of TCR sequences and their cognate epitope.

**Single-cell RNA-sequencing with single-cell immuno-profiling of the sorted CMV-CTLs.** CMV-CTLs at 9-11 months after allo-HCT were sorted in three recipients (5000, 2300, and 1500 cells, respectively). After being sorted into one PCR tube per sample, cells were directly transferred to GENEWIZ JAPAN (Kawaguchi, Japan), and single-cell RNA-sequencing with single-cell immuno-profiling V(D)J analyses was performed according to the manufacturer's instructions. Briefly, the input volume of cells was adjusted as 1000 cells were estimated to be recovered. Individual cell beads in emulsion with single cells (GEMs) were generated with a 10x Genomics Chromium Single Cell A Chip (10x GENOMICS) and Chromium Single Cell 5′ Library & Gel Bead Kit (10x GENOMICS) on a Chromium Controller (10x GENOMICS) followed by RT within each GEMs. cDNA were then amplified and subjected to adapter ligation and PCR amplification, to establish libraries to study gene expression at the single-cell level with a Chromium Single-Cell 5′ Library Construction Kit (10x GENOMICS). Simultaneously, human TCR regions were enriched with specific primers from the cDNA above using a Chromium Single-Cell V(D)J Enrichment Kit, Human T Cell (10x GENOMICS). After library preparation and a quality check, the gene expression and V(D)J libraries were pooled in a ratio of 9:1, and sequencing was performed with a NovaSeq 6000 S4 Reagent Kit (Illumina) on an Illumina NovaSeq sequencer (Illumina). The Chromium single-cell RNA-seq output was processed on Cell Ranger ver3.02 (10x GENOMICS) to align reads and generate matrices, and the results including cloupe and vloupe files were re-transferred to Jichi Medical University for further analyses.

The cloupe and vloupe files obtained above were submitted to Loupe Cell Browser (ver.4.0.0) and Loupe VDJ Browser (ver.3.0.0) for interactive visualization of clustering data on gene expression and VDJ immune-profiling. The dot plots were categorized according to their clonotypes by TCRβ-CDR3 alone. The feature genes among the top 3 clones were compared by a globally distinguishing method.

**RNA-sequencing of the sorted CMV-CTLs.** We analyzed GEP of CMV-CTLs in 37 samples: 21 samples in the early phase ($n = 5$ at 1 month, 6 at 2 months, and 10

at 3 months after allo-HCT), and 16 samples in the late phase ($n = 1$ at 9 months, 12 at 1 year, and 3 beyond 1 year after allo-HCT). Up to 1000 CMV-CTLs (range: 57–1000) were sorted directly into one PCR tube per sample. After sorting, CMV-CTLs were treated by a SMART-Seq® v4 Ultra® Low Input RNA Kit for Sequencing (Takara Bio, Inc.) according to the manufacturer's instruction with a modified lysis components (Supplementary Table 3) and thermal cycles using Veriti (Thermo Fisher Scientific). Briefly, first-strand cDNA was synthesized after direct sorting and cell lysis, and full-length cDNA amplification was performed by Ligation-Dependent PCR with 16 cycles. SPRI purification was then performed using an Agencourt AMPure XP PCR purification kit (Beckman Coulter). Electrophoresis analyses were performed to examine the quality of the amplified cDNA using an Agilent 2100 Bioanalyzer (Agilent Technologies) with High Sensitivity DNA Chips (Agilent Technologies). If a sufficient yield of cDNA was achieved with a distinct peak spanning 400 bp to 10,000 bp (Supplementary Fig. 7b), the synthesized cDNA was transferred to Takara Bio, Inc. (Otsu, Japan). The cDNA libraries for sequencing were prepared with a Nextera XT DNA Library Prep Kit (Illumina) and a Nextera XT Index Kit v2 SetA/B/C/D (Illumina) or IDT for Illumina DNA/RNA UD Indexes, and then sequenced with a NovaSeq 6000 S4 Reagent Kit (Illumina) and a NovaSeq Xp 4-Lane Kit (Illumina) on Illumina NovaSeq sequencer (Illumina). Sequence analyses were performed with NovaSeq Control Software v1.6.0 or v1.7.0, RTA v3.4.4, and bcl2fasq2 v2.20. The mapping, annotation, and calculation of gene expression were performed with DRAGEN Bio-IT Platform v3.6.3 using GRCh38.primary_assembly.genome.fa.gz (GEN-CODE) and gencode.v35.primary_ assembly.annotation.gtf.gz (GENCODE), and the results were re-transferred to Jichi Medical University for further analyses.

**Gene expression profile, PPI network construction, and GO analyses**. The read count data of the gene expression profile obtained from RNA-sequencing were analyzed with the "edgeR" package (ver.3.30.3)[69,70], through Bioconductor (ver.3.12) (https://bioconductor.org/packages/release/bioc/html/edgeR.html) on R (ver.4.03) and EZR (ver1.54) (http://www.jichi.ac.jp/saitama-sct/SaitamaHP.files/statmedEN.html)[66]. The likelihood ratio test was used to test for differential expression, and an ANOVA-like comparison was performed to find the DEGs among the groups in this study. Heatmaps were created by the heatmap.2 function in the "gplots" (ver.3.1.1) package using a matrix of log CPM (counts per million of the DEGs in the compared groups of interest). The bio-statistical approach of enrichment analyses was performed according to previous reports[71,72]. Briefly, the DEGs with a P-value <0.05 in the compared groups of interest were applied to GO enrichment analyses to broadly identify possible biological processes or networks, as long as their FDRs were <0.40. The protein–protein interaction (PPI) network in CMV-CTLs was constructed through STRING (Search Tool for the Retrieval of Interacting Genes/Proteins, ver.11.0, http://string-db.org/), which is a biological database[73]. The component genes of the PPI network were selected with a confidence score of >0.4. The visualized PPI network was modified by Cytoscape (ver.3.8.2, http://www.cytoscape.org/). Next, ClueGO (ver.2.5.8) and CluePedia (ver.1.5.8), which is a Cytoscape plug-in App[74,75], were used to show the pathways with <0.05 of P-value without Bonferroni correction and to visualize the GO terms of the biological and immune system processes in the functionally grouped network of CMV-CTLs. The visualized ontology network was also modified by Cytoscape.

**Reporting summary**. Further information on research design is available in the Nature Research Reporting Summary linked to this article.

## Data availability
All data associated with this study are present in the paper or the Supplementary Information. We provided an original source data for the main figures as Supplementary Data 11 and 12, and explained how to use variables in those Data to plot Figs. 1b, 2a, 2c–f, and 3b–d in Supplementary Data 13. Source data underlying Figs. 4–8 are presented in Supplementary Data 2–10. Gene expression profile data have been deposited in the GEO database under the accession number (GSE156383).

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

## Acknowledgements

The authors are grateful for the contributions of all of the participating donors, patients, physicians, and data managers. H.N. was a recipient of a grant for the Japanese Initiative for Progress of Research on Infectious Disease for Global Epidemic (J-PRIDE) from the Japan Agency of Medical Research and Development (AMED) (JP19fm0208015). Y.K. was a recipient of a grant for Practical Research for Innovative Cancer Control from AMED (JP21ck0106479).

## Author contributions

H.N designed the study, performed the experiments, collected and analyzed data, and wrote the manuscript. M.K. and K.T.-S. performed the experiments. K. Kawamura, Y.A., M.K., J.T., S.K., N.Y., K.Y., Y.M., A.G., K. Kameda, A.T., M.T., S. Kimura, and S. Kako collected samples and clinical data, and revised the manuscript. Y.K. designed the study, advised on the methods, wrote the manuscript, and was responsible for the projects.

## Competing interests

H.N. has received honoraria from Takeda Pharmaceutical, Otsuka Pharmaceutical, Bristol-Myers Squibb, Celgene, Pfizer, Novartis, Janssen Pharmaceutical, Eisai, Chugai Pharmaceutical, and Nippon Shinyaku. S. Kimura has received honoraria from Asahi Kasei, Sumitomo Dainippon Pharma, MSD, Astellas, Pfizer, Kyowa Kirin, Chugai Pharmaceutical Co., Ltd., Bristol-Myers Squibb, Celgene, Ono Pharmaceutical Co., Ltd., Eisai Co., Ltd., and Nippon Kayaku. S. Kako has received honoraria from Novartis, and Bristol-Myers Squibb. Y.K. has received honoraria from MSD, Astellas, Sumitomo Dainippon Pharma, Chugai Pharmaceutical, Pfizer, Novartis, Otsuka Pharmaceutical, Eisai, and Janssen Pharmaceutical Kyowa Kirin, Takeda Pharmaceutical, Ono Pharmaceutical, Shionogi & Co., Bristol-Myers Squibb, Celgene, Mochida, Alexion, and Takara-Bio, and research grants from Celgene, Astellas, Chugai Pharmaceutical, Kyowa Kirin, Ono Pharmaceutical, Sumitomo Dainippon Pharma, Takeda Pharmaceutical, Eisai, Shionogi & Co., Otsuka Pharmaceutical, Nippon-Shinyaku, Taiho, Pfizer, MSD, Asahi-Kasei Corporation, Sanofi, Novartis, Taisho-Toyama, CSL Behring, and Tanabe-Mitsubishi. The other authors report no potential competing conflicts of interest.
