## [Transparent Peer Review File · Communications Biology]

Reviewers' comments:

Reviewer #1 (Remarks to the Author):

General remarks:

Adoptive T cell therapy targeting CMV has been one of the pioneering 'use cases' for (and thus still represents one of the best settings to study) adoptive T cell therapy in general. The value of the manuscript therefore primarily lies in an in-depth profiling of clinically transferred T cells, rather than being "the first report (...) on HLA-A24 CMV CTLs". In other words, the manuscript would not provide much of an impact if it just studied yet another HLA-directed response, but rather addresses features in a specific setting (i.e. HLA-A24 CMV CTLs) that could be relevant for T cell therapy in general. The general research question of the manuscript is therefore relevant. Also, the clinical cohort with quite a few patients and – importantly – categorization into different time points and virus reactivation groups (no reactivation, one reactivation, multiple reactivation) represents a valuable resource to address important questions. Unfortunately, the manuscript is not very well written, the figures are not clear in their presentation, main findings are not interpreted in a comprehensive manner and there are several fundamental limitations which are not discussed or dealt with. The manuscript therefore is not able to exploit the resource that the data could provide.

Major points:

- Clarity: The text, the figures and the figure legends are lacking partly very important information. For example, the exact composition of the patient groups for individual analyses is not clearly presented or figures are not legible and lack important information with regards to what can be seen. See specific points below for further examples.
- CDR3 sequence (motif) analyses: First, a major limitation (which is not mentioned by the authors) of the study is that only CDR3 beta sequences are evaluated, whereas the TCR is a heterodimeric receptor consisting of an alpha and a beta chain. This has important implications for nearly all of the findings of the manuscript. Second, the sequence analyses are merely observational, without any form of functional validation. This concerns in particular the binding prediction. Given the fact that the data are only observational, the findings are not contextualized well enough in reference to hypotheses and literature, and also their implications for an enhanced understanding of the CMV T cell response or immunotherapy remains completely elusive.
- Repertoire diversity analyses: Fig. 4 in particular lacks important information and is not interpreted clearly. Fig. S3 conveys a much better idea of what would actually be possible in terms of analyses with the data at hand. This epitomizes the opportunities the authors would have with the data, but which they do not exploit with the manuscript in its present form.
- Single-cell RNA sequencing: A homogenous GEP distribution among individual clones in a recipient is completely expected given the fact that cells with a given antigen specificity have been sorted. This has also in principle been shown in many studies before and is therefore not a novel finding.
- GEP, PPI network, GO enrichment analyses: Again, given the fact that the data are only observational, the findings are not contextualized well enough in reference to hypotheses and literature, and also their implications for an enhanced understanding of the CMV T cell response or immunotherapy remains completely elusive.

Specific points*:

- it is called "adoptive" T cell therapy (not adaptive; T cells are adopted by the host organism)
- in addition to refs 18-21, more important studies showing proof-of-concept of CMV-specific T cell therapy should be mentioned, particularly PMIDs 1352912, 7675046, 16061727, 28090089
- Fig. 1 should be moved to the supplementary figures and be mentioned after the general set-up (i.e. how many CTLs from how many samples from how many patients were analyzed etc)
- More detailed information on the patients needs to be clearly visualized in the first main figure and also be more explicitly explained in the text beyond the mere indication to Dataset S1 (i.e. How many cells were analyzed for each patient? How do the patients distribute across the reactivation groups, i.e. no reactivation, one episode, repeated. How are early phases defined etc. These important

information are so far only hidden in the methods and in the supplementary data)

- Fig. 2a: Why is BV7, if it is the most frequent TRBV chain observed, not displayed in the 3D plot? The same occurs for BJ2-1 and 2-3. In general, the 3D plot does not seem to convey any additional help for interpretation beyond the Circos plot in Fig. 2b and should be therefore removed.

- Fig. 2d: In order to be better able to evaluate the data, the number of clones and number of patients that contributed to each plot should be indicated in the figure legend; the same applies also to other sub-figures, e.g. Fig. 3 in which it is not clear what $n=x$ is referring to. In general, the manuscript lacks specific information on many occasions, for example also in line 123 (major clones accounting for $\geq 5\%$ of what?)

- Lines 119-122: Why do the authors hypothesize that clones with GGG in ≤ 15 AA or GG in ≥ 16 AA or clones in the no-CMV reactivation group show a high binding affinity? pp65 is rather expressed in the acute phase of the infection (and not during latency, for which a reverse repertoire evolution towards dominance of low affinity clones has been recently shown, PMID 32205883). So, I would hypothesize that after repeated infection, avidity maturation should take place, with preferential selection of high affinity clones. This actually seems to be supported by the data (Fig. 3F), even though the differences seem to be not statistically significant. Whether clones with GGG in ≤ 15 AA or GG in ≥ 16 AA would show high or low affinity I would not have been able to predict. I understand that those were features that were disproportionately often found in the CTLs. However, they were not functionally selected after repeated infections (see argument above). In general, binding predictions based on CDR3 beta alone are only a weak indicator of TCR affinity. First, TCR affinity is dependent on both CDR3 beta and alpha. Second, to my knowledge no experimental/functional data exist that show that TCR binding predictions are able to predict TCR affinity well.

- In this regard, the predicted loop structures of Fig. 3G are aesthetically nice, but their contribution to an enhanced understanding is questionable. How is the statement "seemed to have a relatively symmetric groove" (line 130) supported by the data? What kind of predictions could be made based on the predicted loop structures? Can those predictions then be tested/validated? This is just one of many examples in the manuscript where qualitative observations are not quantitatively/functionally validated.

- Fig. 4 is difficult to understand with the information available. Fig. 4D/E: It is not clear, whether early-late comparisons were done for all reactivation groups pooled. Because if they were, this could average out differences. What would this look like for the individual reactivation groups? Fig. 4F: It is not clear, whether samples (bars in a given order) represent longitudinal samples (early vs. late) in the same patients. I assume each color annotates one clone (this is neither stated in the legend nor in the figure itself). Also, the figure description in the text is not concise et al. In Fig. S3 (which is unfortunately in the supplementary), it is much clearer, because it is clear that these are longitudinal observations, the important information with the clone distribution in both donor and recipient is given, it is clear which reactivation group these data belong to and which time points are shown. Visualizations like this should be implemented for the other reactivation groups as well. Overall it is not surprising that the repertoires remain stable from early to late in the no reactivation group and that they change upon single or repeated reactivations. What is surprising is that the repertoire diversity is higher in the early phase of the repeated CMV reactivation, because one would expect repertoire focusing to have taken place. Furthermore, potential reasons for a more diverse repertoire at late compared to early time points are not discussed/tested. Could this be due to newly formed T cells originating from the transferred HSCs? The authors should describe and discuss this more clearly, also in the context of the many studies on TCR repertoire diversity during CMV infection that exist.

- Fig. 5A-C: The figures should be labelled with TSNE-1 and TSNE-2. Also, in my eyes the numbers for the clones refer to clone counts (not proportions, as it is stated in the figure legend). In Fig. 5D-F, if the upper label is supposed to be gene names, it is useless since it is not legible. For the figure to be interpretable, sub-figures D-F should be much larger. These are just few of many examples of sloppy annotations of figures and legends as well as sloppy figure design (legibility, sub-figure sizes, font sizes, font styles etc etc) throughout the whole manuscript.

- Fig. 6A: It is nice to see how consistently the patients cluster according to the clinical group. However, again, gene names are not eligible and the sub-figure should be larger.

- lines 198-200: references for the individual genes should be included; also, it is not clear how "degrees of centrality and read counts" (line 197) were quantitatively used as cut-offs / filters.
- headers in lines 191, 205 and 214 have (accidentally?) circles before the headers
- Fig. 7, S4 and S5: See comments above regarding figure design: Illegible gene names, sloppy annotations, incomplete legends (e.g. degree of what? in Fig. 7, although this is mentioned in the legends). Importantly, does a high log₂fc indicate upregulation in CMV-positive individuals? A discussion/interpretation or at least an attempt of contextualization of these findings is missing in the manuscript.

*This list of specific points is not exhaustive, this means that other points could have been raised in addition. However, the review in its current form should indicate the general line of criticism well enough.

Reviewer #3 (Remarks to the Author):

This is a very interesting, data-rich but also data-dense report evaluating both T-cell receptor diversity and gene expression profiles of HLA A2402 and A2420 restricted T-cells specific for QYD peptide of CMVpp65 presented by these alleles, in CMV-seropositive recipients of allogeneic hematopoietic cell transplants who did or did not have 1 or more reactivations of CMV viremia post transplant. The data presented are extensive and of considerable interest. However, the hypotheses to explain the data are tentative and limited in scope, thereby also limiting interpretation of the data presented. Certain aspects of the data presented need to be further described and clarified. Given the amount of data presented, it would also be worthwhile to strengthen the discussion and better identify central hypotheses and the implications of the work presented. Thereeto, the following modifications should be considered:

1) Table 1 is instructive but raises concerns regarding the results presented unless certain data regarding the patients is integrated into the data presented in the figures. This is particularly the case for the data presented regarding the TCRs and T-cells derived from patients at specific times post transplant. Specifically:

a. 9 of the 26 patients received transplants from HLA disparate donors. Did all recipients and donors share HLA 2402 and A 2420?

b. Was the donor's, or the post transplant recipient's T-cell response to the QYD peptide presented by HLA 2402 the dominant response?

The concern here is that if the recipient does not share HLA 2402, the donor's T-cells would not recognize the QYD peptide and would fall off in frequency. Similarly, if the donor inherits HLA B0702 which is always dominant, the proportion of tetramer+ cells would be reduced. This could contribute to the lower proportion of Tet+ cells in patients with recurrent CMV viremia.

2) Were the descriptions of the TCR sequences and their binding affinities in patients early and late post transplant based only on patients who received transplants from seropositive donors? If this is the case, it would be useful to state this up front, it would greatly clarify interpretation of data. If T-cells generated post transplant from seronegative donors are also included, were their TCR sequences shorter, or lacking in GGG sequences, or of lower affinity?

3) The data presented in Figure 3 provide evidence that the CDR3 amino acid sequences for patients who don't develop CMV reactivations are longer, and are more likely to contain GGG motifs. Those longer TCRs with GGG motifs are also of higher affinity with more symmetrical structures potentially better able to bind peptide HLA complexes. These data also support the central hypothesis of the next section which is that patients who don't experience a CMV reactivation, or only have 1 such reactivation have received transplants from seropositive donors whose dominant CMVpp65 specific T-cells are better able to confer resistance on the transplanted host. Against this hypothesis, however, is Figure 3F, which suggests that CMVpp65 T-cells that are A24 restricted and have TCRs that exhibit

GGG motifs and presumably higher affinity are present at frequencies that are not different in both the no reactivation and the multiple CMV viremia episode groups. The authors provide no explanation for Figure 3F, bringing into question whether any of the data on the TCRs has clinical significance. It is also not considered in the Discussion. However, there are many possibilities that could explain Figure 3F, both in terms of the T-cells (e.g. actual number of the T-cells in the blood, level of expression of costimulatory molecules; effector cytokines such as TNF or IFN γ , or the T-cell cytotoxic activity) or properties of the patient's endogenous virus (e.g. evasins, other factors preventing antigen presentation). This needs to be addressed.

4) The section on the changes in proportion and diversity of the A24 restricted CMVpp65-specific T-cells is of interest in that it provides evidence that dominant CMV T-cells in the donor are also dominant in the host. The data also suggest that there is the expected diversification of T-cells late post HCT, but that while the dominant clone usually persists, it falls off in patients with recurrent infections. Again, the data presented only address proportions of cells, no actual numbers. This needs to be clarified since the number of antigen-specific T-cells is essential to the response. Also, neither the results section nor the Discussion provide the author's thoughts as to why the dominant clones are lost, nor an explanation for the absence of a proportionate rise in new clones to address recurrent infection.

5) The data presented on the gene expansion profiles of the T-cells are a major and novel feature of the report. The data comparing these profiles in CMV T-cells generated from naïve donors vs seropositive donors are particularly important. Although the authors have previously presented data regarding recipients of grafts from CMV seropositive donors who experience CMV reactivation, its absence from this report as a frame of reference limits comparative analysis of the gene expression profiles of the T-cells generated from naïve seronegative donors.

The data presented on the profiles late post transplant are of particular interest, and deserve further discussion, since the activation of genes for CD28 and other costimulatory molecules in the No-P group may reflect their key role in enhancing the effector function of lower avidity T-cells. Similarly, the genes activated in the T-cells from individuals with repeated episodes of CMV viremia suggest that the T-cells themselves may be ineffective against the host CMV but are capable of signaling NK cells, B cells and components of the innate immune system.

The Discussion needs significant revision. As it is, it reiterates the findings in the results but does not put those data in context for comparison with data presented in prior reports. The discussion of the potential significance of CD28 as a survival signal is worthwhile. However, the contribution of CD28 and other costimulatory factors to the function of T-cells with lower affinity TCRs should also be considered, since the function of T-cells with a diversity of TCR affinity is an important finding in this report. Similarly, the discussion of the genes expressed by T-cells from individuals with recurrent CMV needs to be further developed. There is a wealth of information presented here but it needs to be clarified and more extensively explored as a door to the future rather than a fine tuning of the past.

To Reviewer #1:

Reviewer comments:

Major points:

1) Clarity: The text, the figures and the figure legends are lacking partly very important information. For example, the exact composition of the patient groups for individual analyses is not clearly presented or figures are not legible and lack important information with regards to what can be seen. See specific points below for further examples.

- Author response:

We really appreciate the reviewer's suggestion. We modified the tables, figures and corresponding figure legends as follows.

Figure legends:

“Fig. 1. Preference in the selection of complementarity-determining region 3 (CDR3) of T cell receptor (TCR)- β of all identified HLA-A24-restricted CMV-pp65-specific cytotoxic T-cell (CMV-CTL) clones. (A) A circos plot showing a skewed preference in the selection of BV and BJ genes of all CMV-CTL clones. (B) A column chart for the amino acid (AA) length of all CMV-CTL clones. (C) AA sequence logos in all CMV-CTLs observed after allo-HCT with 17, 16, 15, or 14 AA-long TCR β -CDR3, and logos in the groups divided according to AA length and CMV reactivation pattern (n=10 in the no-CMV reactivation group, n=9 in the one-episode group, and n=7 in the repeated-CMV reactivation group). X-axis denotes the position of amino acids from the N- to C- terminal.”

“Fig. 2. Amino acid (AA) sequence logos and AA length in the major HLA-A24-restricted CMV-pp65-specific cytotoxic T-cell (CMV-CTL) clones accounting for >5% of all CMV-CTLs within individual recipients in the early or late phases after allo-HCT. (A) AA length according to CMV-reactivation pattern: no-CMV reactivation (n=10), one-episode (n=9), and repeated reactivation (n=7) groups. The AA length of TCR β -CDR3 decreased in the following order: no-CMV reactivation > one-episode (n=9) > repeated CMV reactivation groups (P=0.015 by the

Jonckheere-Terpstra (J-T) test for decreasing tendency). (B) AA sequence logos of TCR β -CDR3 in the major CMV-CTLs with 15 AA-long CDR3-TCR β according to the CMV-reactivation pattern. (C) Difference in TCR-peptide binding scores among the major clones according to AA length of CDR3-TCR β (P=0.003 by the Kruskal-Wallis (K-W) test). (D) Difference in TCR-peptide binding scores among the major clones according to the presence of “(G)GG” motifs in CDR3-TCR β (P<0.001 by the Mann-Whitney U (M-W) test). (E) Difference in TCR-peptide binding scores among the major clones according to the CMV-reactivation pattern (P=0.31 by the K-W test). (F) Difference in binding score weighted by the individual clone counts among the major clones in the early phase of allo-HCT according to the CMV-reactivation pattern (P=0.02 by the K-W test, and P=0.0031 by the J-T test for decreasing tendency). “

“Fig. 3. Predicted structures of 15 amino acid (AA)-long TCR β -CDR3 with the “GGGG” motif and higher binding scores, compared to those with lower binding scores. Their predicted structures are shown with and without a molecular surface model (left and right, respectively). “

“Fig. 4. Changes in proportions and diversity of all HLA-A24-restricted CMV-pp65-specific cytotoxic T-cell (CMV-CTL) clones. (A) Correlation between the proportions of CMV-CTLs to CD8 T-cells and Shannon’s equitability index in the early phase of allo-HCT (P=0.56 by Pearson’s correlation test). (B) Proportions of CMV-CTLs in the early phase of allo-HCT according to the CMV-reactivation pattern: no-CMV reactivation (n=10), one-episode (n=9), and repeated reactivation (n=7) groups (P=0.68 by the Kruskal-Wallis (K-W) test, and P=0.19 by the Jonckheere-Terpstra (J-T) test for decreasing tendency). (C) Shannon’s equitability index in the early phase of allo-HCT according to the CMV-reactivation pattern (P=0.77 by the K-W test, and P=0.24 by the J-T test for increasing tendency). (D) Changes in the proportions of individual CMV-CTL clones between the early and late phases after allo-HCT in each patient for whom samples were available at both points (n=22). The same colors suggest the same CMV-CTL clones between the phases within individual patients. (E) Changes in the proportions of CMV-CTLs

when all groups were pooled between the early and late phases of allo-HCT ($P < 0.001$ by Wilcoxon's signed rank test). (F) Changes in Shannon's equitability index of CMV-CTLs in all groups pooled between the early and late phases of allo-HCT ($P = 0.046$ by Wilcoxon's signed rank test). (G) Difference in TCR-binding scores of the dominant CMV-CTLs, which accounted for $> 35\%$ of all CMV-CTLs within individual recipients in the early or late phases after allo-HCT, between the decreasing and increasing clones at the late phases ($P = 0.016$ by the Mann-Whitney test).”

“Fig. 5. Single-cell RNA-sequencing of HLA-A24-restricted CMV-pp65-specific cytotoxic T-cells (CMV-CTL). (A-C) t-SNE plots and clone counts in (A) Case14, (B) Case4, and (C) Case3. The input volume of cells for sequencing was adjusted as 1,000 cells were estimated to be recovered. Each dot represents a single cell and each color denotes the same individual clone. There was no robustly significant difference with a two-fold change and $P < 0.05$ in gene expression among the clones in each recipient, except for the TCR genes in all cases and GNLV and FXYD5 in Case3. The t-SNE clustering plots are shown after deleting apparently dying cells from Case3.”

“Fig. 6. Gene expression profile (GEP), protein-protein interaction (PPI) network, and gene ontology (GO) enrichment analyses in HLA-A24-restricted CMV-pp65-specific cytotoxic T-cells (CMV-CTL) according to the CMV reactivation pattern in the early phase of allo-HCT. (A) A clustering heatmap of GEP using the top 100 differentially expressed genes (DEGs) according to the CMV-reactivation pattern: no-CMV reactivation (no-group, $n=8$), one-episode of CMV reactivation (one-group, $n=7$), and repeated episodes of CMV reactivation (repeated-group, $n=6$). (B) PPI network constructed by the top 100 DEGs which had close connections with each other. The shape size suggests the degree of centrality of the PPI network, meaning how many edges are connected to each gene node. The heat color denotes the log-fold changes in individual gene expression between the no- vs. repeated-groups (left) or between the one- vs. repeated groups (right). Only the networks with ≥ 5 connections are shown. (C) GO and the shared genes derived from the top

200 DEGs. Each circle denotes an identified term with a P-value of <0.05 without the Bonferroni correction. The same colors mean GO terms that belong to the same GO term-tree groups. Only the names of the leading GO terms with highest significance in each GO term-tree group are shown.”

“Fig. 7. Gene expression profile (GEP), protein-protein interaction (PPI) network, and gene ontology (GO) enrichment analyses in HLA-A24-restricted CMV-pp65-specific cytotoxic T-cells (CMV-CTL) in the early phase of allo-HCT according to the donor CMV serostatus. (A) A clustering heatmap of GEP using the top 100 differentially expressed genes (DEGs) between CMV-seronegative (n=11) and -seropositive (n=10) donors. (B) PPI network constructed from the top 100 DEGs which had close connections with each other. The shape size suggests the degree of centrality of the PPI network, meaning how many edges are connected to each gene node. The color heat denotes log-fold changes in individual gene expression between the CMV-seropositive vs. -seronegative donor groups. Only networks with ≥ 5 connections are shown. (C) GO and the shared genes derived from the top 200 DEGs. Each circle denotes an identified term with a P-value of <0.05 without the Bonferroni correction. The same colors mean GO terms that belong to the same GO term-tree groups. Only the names of the leading GO terms with the highest significance in each GO term-tree group are shown.”

“Fig. 8. Gene ontology (GO) enrichment analyses focusing on the immune system processes in HLA-A24-restricted CMV-pp65-specific cytotoxic T-cells (CMV-CTL) in the early phase of allo-HCT according to the CMV reactivation pattern in the subgroups of donor CMV serostatus. (A) GO of the immune system processes and the shared genes derived from the top 500 differentially expressed genes (DEGs) with $FDR < 0.1$, focusing on the CMV-seropositive cohort alone. Each circle denotes an identified term with a P-value of <0.05 without the Bonferroni correction. The fused GO terms are shown. The same colors mean GO terms that belong to the same GO term-tree groups. The names of the leading GO terms are shown, based on the number of genes in each GO term-tree group. (B) The log-fold changes in the shared genes in the no-CMV (no-group, n=5) and one-episode (one-

group, n=3) reactivation groups compared with the repeated CMV reactivation group (repeated group, n=2). (C) GO of the immune system processes and the shared genes derived from the top 324 DEGs with FDR<0.4, focusing on the CMV-seronegative cohort alone. Each circle denotes an identified term with a P-value of <0.05 without the Bonferroni correction. The fused GO terms are shown. The same colors mean GO terms that belong to the same GO term-tree groups. The names of the leading GO terms are shown based on the number of genes in each GO term-tree group in addition to the term of “T-cell costimulation”. (D) The log-fold changes in the shared genes in the no-CMV (no-group, n=3) and one-episode (one-group, n=4) reactivation groups compared with the repeated CMV reactivation group (repeated group, n=4). “

“Fig. 9. Gene expression profile (GEP), protein-protein interaction (PPI) network, and gene ontology (GO) enrichment analyses in HLA-A24-restricted CMV-pp65-specific cytotoxic T-cells (CMV-CTL) according to the CMV reactivation pattern in the late phase of allo-HCT. (A) A clustering heatmap of GEP using the top 100 differentially expressed genes (DEGs) according to the CMV-reactivation pattern: no-CMV reactivation (no-group, n=6), one-episode of CMV reactivation (one-group, n=6), and repeated episodes of CMV reactivation (repeated-group, n=4). (B) PPI network constructed by the top 100 DEGs which had close connections with each other. The shape size suggests the degree of centrality of the PPI network, meaning how many edges are connected to each gene node. The color heat denotes the log-fold changes in individual gene expression between the no- vs. repeated-groups (left) or between the one- vs. repeated groups (right). Only networks with ≥ 5 connections are shown. (C) GO and the shared genes derived from the top 200 DEGs. Each circle denotes an identified term with a P-value of <0.05 without the Bonferroni correction. The same colors mean GO terms that belong to the same GO term-tree groups. Only the names of the leading GO terms with the highest significance in each GO term-tree group are shown. (D) The log-fold changes in the shared genes in the no- and one-groups compared with the repeated group.”

Supplementary Figure legends:

“Fig. S1. A simple scheme for identification of HLA-A24-restricted CMV-pp65-specific cytotoxic T-cells (CMV-CTLs) and correlation of the current next-generation sequence (NGS) strategy and our previous direct single-cell method. (A) Identification of CMV-CTLs and difference in sorting between the methods. CMV-CTLs were defined as CD3+CD8+HLA-A*24-CMV-pp65 (QYDPVAALF)-tetramer+ T-cells. (B) Correlation between individual clone proportions identified by the current NGS strategy and our previous direct single-cell method ^{9, 10} ($r=0.971$, $P<0.001$ by Pearson’s correlation test, $n=5$ samples).”

“Fig. S2. Amino acid (AA) sequence logos of the top 1 or major clones. (A) AA sequence logos of the top 1 clones of HLA-A24-restricted CMV-pp65-specific cytotoxic T-cells (CMV-CTL). Logos of 15 and 14 AA of TCR β -CDR3 are shown, but there were too few top 1 clones with >15 or <14 AA to create sequence logos. (B) AA sequence logos of TCR β -CDR3 according to AA length in major CMV-CTLs accounting for $>5\%$ of all CMV-CTLs within individual recipients in the early or late phases after allo-HCT. X-axis denotes the position of amino acids from the N- to C- terminal.”

“Fig. S3. TCR-peptide binding scores in all of HLA-A24-restricted CMV-pp65-specific cytotoxic T-cell (CMV-CTL) clones. (A) TCR-peptide binding scores according to amino acid (AA) length ($P<0.001$ by the Kruskal-Wallis (K-W) test). (B) TCR-peptide binding scores between the CMV-CTL clones with and without the “GGG” motif in ≤ 15 AA or “GG” motif in ≥ 16 AA in their TCR β -CDR3 ($P<0.001$ by the Mann-Whitney U (M-W) test). (C) AA sequence logos in CMV-CTLs with higher and lower binding scores to the CMVpp65 peptide according to AA length of CDR3-TCR β . A score higher than a median value of 0.141 was defined as a higher binding score.”

“Fig. S4. Time-dependent changes in HLA-A24-restricted CMV-pp65-specific cytotoxic T-cells (CMV-CTLs) within individual patients between the early and late phases of allo-HCT. (A) Changes in proportions of CMV-CTLs between the

early and late phases of allo-HCT according to the CMV reactivation pattern: no-CMV reactivation (n=9), one-episode (n=6), and repeated reactivation (n=7) groups. (B) Changes in Shannon's equitability index of CMV-CTLs between the early and late phases of allo-HCT according to the CMV reactivation pattern. (C) The proportions of CMV-CTL clones in the donor, one month after allo-HCT, and >one year after allo-HCT in the no-CMV reactivation group with CMV-seropositive donors (Cases 1, 2, & 3). (D) Changes in the proportions of the top 1 clones of CMV-CTLs within individual patients between the early and late phases of allo-HCT (P=0.48 by the Jonckheere-Terpstra (J-T) test for decreasing tendency). (E) The proportions of CMV-CTL clones that were observed in the early phase but disappeared in the late phase of allo-HCT. Their proportions increased in the following order: no-CMV reactivation < one-episode < repeated CMV reactivation groups (P=0.026 by the J-T test for increasing tendency). (D) The proportions of the newly-appeared CMV-CTL clones in the late phase of allo-HCT. No significant difference was observed according to the CMV-reactivation pattern (P=0.40 by the J-T test for increasing tendency)."

"Fig. S5. Gene expression profile (GEP), protein-protein interaction (PPI) network, and gene ontology (GO) enrichment analyses in HLA-A24-restricted CMV-pp65-specific cytotoxic T-cells (CMV-CTL) according to the CMV reactivation pattern in the early phase of allo-HCT, focusing on the CMV-seropositive donor cohort. (A) A clustering heatmap of GEP using the top 100 differentially expressed genes (DEGs) according to the CMV-reactivation pattern: no-CMV reactivation (no-P group, n=5), one-episode of CMV reactivation (one-P group, n=3), and repeated episodes of CMV reactivation (repeated-P group, n=2). (B) PPI network constructed by the top 200 DEGs with FDR of <0.05 which had close connections with each other. The shape size suggests the degree of centrality of the PPI network, meaning how many edges are connected to each gene node. The color heat denotes the log-fold changes of the individual gene expression between the no-P vs. repeated-P groups (left) or between the one-P vs. repeated-P groups (right). Only networks with ≥ 5 connections are shown. (C) GO and the shared genes derived from the top 200 DEGs. Each circle denotes an identified term with a P-value of

<0.05 without the Bonferroni correction. The same colors mean GO terms that belong to the same GO term-tree groups. Only the names of the leading GO terms with the highest significance in each GO term-tree group are shown.”

“Fig. S6. Gene expression profile (GEP), protein-protein interaction (PPI) network, and gene ontology (GO) enrichment analyses in HLA-A24-restricted CMV-pp65-specific cytotoxic T-cells (CMV-CTL) according to the CMV reactivation pattern in the early phase of allo-HCT, focusing on the CMV-seronegative donor cohort. (A) A clustering heatmap of GEP using the top 100 differentially expressed genes (DEGs) according to the CMV-reactivation pattern: no-CMV reactivation (no-N group, n=3), one-episode of CMV reactivation (one-N group, n=4), and repeated episodes of CMV reactivation (repeated-N group, n=4). (B) PPI network constructed by the top 120 DEGs with FDR of <0.05 which had close connections with each other. The shape size suggests the degree of centrality of the PPI network, meaning how many edges are connected to each gene node. The color heat denotes log-fold changes in the individual gene expression between the no-P vs. repeated-P groups (left) or between the one-P vs. repeated-P groups (right). Only networks with ≥ 5 connections are shown. (C) GO and the shared genes derived from the top 200 DEGs. Each circle denotes an identified term with a P-value of <0.05 without the Bonferroni correction. The same colors mean GO terms that belong to the same GO term-tree groups. Only the names of the leading GO terms with the highest significance in each GO term-tree group are shown.”

2) CDR3 sequence (motif) analyses: First, a major limitation (which is not mentioned by the authors) of the study is that only CDR3 beta sequences are evaluated, whereas the TCR is a heterodimeric receptor consisting of an alpha and a beta chain. This has important implications for nearly all of the findings of the manuscript. Second, the sequence analyses are merely observational, without any form of functional validation. This concerns in particular the binding prediction. Given the fact that the data are only observational, the findings are not contextualized well enough in reference to hypotheses and literature, and also their implications for an enhanced understanding of the CMV T

cell response or immunotherapy remains completely elusive.

- Author response:

We totally agree with the reviewer's opinion about the limitation in the analyses of only CDR3 beta and its observational nature. Therefore, we modified the limitation section on pg.23 as follows:

“This study had several limitations. First, it might be too small to draw any definitive conclusions regarding the association of the repertoire diversity with clinical outcomes. Second, we focused on only HLA-A24-restricted CMV-CTLs. Other class-I HLA-restricted CMV-CTLs could also play a role against CMV, which might have contributed to the lower proportion of tetramer-positive cells in patients with recurrent CMV viremia. We must assess the overall repertoire diversity beyond HLA restriction to clarify clinical aspects in the future. Third, we analyzed only TCR β -CDR3, and not TCR α . In addition, the predicted loop structure of TCR β -CDR3 was not validated by X-ray crystallography. Therefore, the actual binding affinity might be different from the predicted score in each T-cell. Our study was based on observational facts. Functional analyses would be warranted in the future, if all CMV-CTL clones could be established and all pairs of TCR α and TCR β could be linked in individual clones. However, this study primarily sought to provide an in-depth profiling of clinically transferred T cells, and is the first to demonstrate the features of GEP in CMV-CTLs following allo-HCT according to the CMV-reactivation pattern as well as those of AA motifs used in HLA-A24 CMV-CTLs. Thus, these results could contribute to the expansion of public databases, leading to future investigations.”

In addition, we added the following on pg.19 of the Discussion section;

“the predicted loop structure of 15 AA-long TCR β -CDR3 with “GGG” motifs seemed like a simple scoop, while that with lower scores seemed more distorted, suggesting that the ease of physical contact is a critical factor for binding between T-cells and peptides. In general, CTLs with high affinity toward antigens

demonstrate an increased cytotoxic function.^{53,54,56,57} CMV-CTLs with a higher binding score could be candidates for adoptive T-cell therapies for refractory CMV diseases, although the actual binding affinity of whole TCR-peptide-MHC complex and its specific cytotoxicity should be confirmed after individual clones are established.”

3) Repertoire diversity analyses: Fig. 4 in particular lacks important information and is not interpreted clearly. Fig. S3 conveys a much better idea of what would actually be possible in terms of analyses with the data at hand. This epitomizes the opportunities the authors would have with the data, but which they do not exploit with the manuscript in its present form.

- Author response:

Thank you for this advice to show figures using the format in Fig.S3. Accordingly, we made figures to show the proportional changes of individual clones between the early and late phases after allo-HCT in each recipient and showed them as Fig.4D. Instead, several figures are provided as supplementary figures (current Fig.S4A-F). In addition, we modified the corresponding section of the Results on pg.9 as follows:

“Focusing on time-dependent changes (Figure 4D), the proportions of CMV-CTLs decreased in the late phases ($P < 0.01$, Fig 4E) and their diversity increased ($P = 0.046$, Fig. 4F) in all groups pooled. However, the statistical significance tended to diminish if we checked the time-dependent changes according to the CMV reactivation pattern (Fig. S4A and S4B). If we consider the estimated cell counts, there were no significant time-dependent changes according to the CMV reactivation pattern.

If we focus on the three pairs of donor and recipient in the no-CMV reactivation group transplanted from CMV-seropositive donors, the dominant clones of each donor remained dominant after allo-HCT (Fig. S4C), suggesting that CMV-CTLs transferred from the donors kept playing an anti-viral role in the cohort. Similarly,

if we consider the most dominant top 1 clones of individual patients in all groups, there was no significant time-dependent proportional change according to the CMV-reactivation pattern between the early and late phases after allo-HCT (Fig. S4D). This meant that the dominant clone in the early phases after allo-HCT usually remained dominant. However, in several cases, some dominant clones decreased in the late phases after allo-HCT (Fig. 4D). Next, we focused on the dominant clones that accounted for > 35 % of all CMV-CTLs within individual recipients in the early or late phases after allo-HCT, and checked the difference in TCR-binding scores between the decreasing and increasing clones in the late phases. The binding scores of the increasing dominant clones in the late phases were significantly higher than those of the decreasing ones (P=0.016, Fig. 4G).

The top 1 and 2 clones at the early phase of allo-HCT accounted for >75% in all of the no-CMV reactivation group, 78% of the one-episode group, and 43% of the repeated CMV reactivation group (P=0.018 by Fisher's exact test)."

In addition, we modified the figure legend as mentioned above.

4) Single-cell RNA sequencing: A homogenous GEP distribution among individual clones in a recipient is completely expected given the fact that cells with a given antigen specificity have been sorted. This has also in principle been shown in many studies before and is therefore not a novel finding.

- Author response:

According to the reviewer's suggestion, we deleted the following sentence from the Discussion section: "this is the first report to show the relatively homogenous GEP distribution of individual CMV-CTL clones within each recipient."

5) GEP, PPI network, GO enrichment analyses: Again, given the fact that the data are only observational, the findings are not contextualized well enough in reference to hypotheses and literature, and also their implications for an enhanced understanding of the CMV T cell response or immunotherapy remains completely elusive.

- Author response:

Thank you for this suggestion. We completely changed the GEP analyses, since the other reviewer requested that we should add more samples. In addition, we re-analyzed all RNA-seq data, and the mapping, annotation, and calculation of gene expression were re-performed with DRAGEN using GRCh38.primary_assembly|genome, instead of the previous platform that used Genedata Profiler Genome, STAR, and GRCh37. Thus, the additional analyses enabled us to simplify the comparisons, and we modified the Results and Discussion accordingly, as follows:

In the Results section:

“GEP, protein-protein interaction (PPI) network, and gene ontology (GO) enrichment analyses of HLA-A24-restricted CMVpp65-specific CTLs in the early phase

- ***Comparison according to the CMV-reactivation pattern***

The explanation of individual genes described below was based on GeneCards (The Human Gene Database, <https://www.genecards.org/>). The top 200 differentially expressed genes (DEGs) with $P < 0.05$ are shown in Dataset S3a. They were well-clustered by the top 100 DEGs (Fig. 6A). Their PPI network was constructed using the top 100 DEGs (Fig.6B). According to their degrees of centrality, meaning how many edges are connected to each gene node, TNF (Tumor Necrosis Factor, a multifunctional proinflammatory cytokine) and HELLS (Helicase, Lymphoid Specific, a lymphoid specific helicase and regulator for the expansion or survival of lymphoid cells)²⁷ seemed to centrally work as hub genes in the network, as well as DNA repair proteins like RAD51AP1 (RAD51 Associated Protein 1, a structure-specific DNA-binding protein involved in DNA repair)²⁸ (Fig. 6B). In addition, TNF was further increased in both the no-CMV reactivation (n=8) and one-episode (n=7) groups compared with the repeated CMV reactivation group (n=6) (Fig. 6B). The GO enrichment analysis demonstrated that the top 200 DEGs were involved in the “regulation of acute inflammatory response” and “regulation of viral genome replication” as an immunological response (Fig. 6C). In these processes,

inflammatory genes such as TNF, and ISG20 (Interferon Stimulated Exonuclease Gene 20, an interferon-induced antiviral exoribonuclease)²⁹ were shared among the biological terms. In addition, they were also involved in the processes of transforming growth factor (TGF)- β pathways, regulation of cell division, and other metabolic activities (Fig. 6C). In summary, TNF expression was increased in the no-CMV reactivation group, and cell replication /division would be more active, leading to the proliferation of CMV-CTLs in that group.

● ***Comparison according to donor CMV-serostatus***

The top 200 DEGs are shown in Dataset S3b. They were also well-clustered by the top 100 DEGs (Fig. 7A). The PPI network suggested that the increased genes of CCR5 (C-C Motif Chemokine Receptor 5, a regulator of granulocytic lineage proliferation),³⁰ LY96 (Lymphocyte Antigen 96, a protein that is associated with toll-like receptor 4 on the cell surface and confers responsiveness to lipopolysaccharide),³¹ S100A8 and S100A9 (S100 Calcium Binding Protein A8 and A9, regulators of inflammatory processes and immune response)³² centrally worked in the CMV-seropositive donor group (n=10) (Fig. 7B). On the other hand, RNA helicases like DDX5 (DEAD-Box Helicase 5, an RNA helicase)³³ were increased in the CMV-seronegative donor group (n=11), suggesting that mRNA processing was more active in this group. The top 200 DEGs were involved in “cellular defense response”, “positive regulation of lymphocyte proliferation”, and “production of molecular mediator involved in inflammatory response” as immunological processes (Fig. 7C). In these processes, inflammatory genes such as CCR5, LY96, LGALS3 (Galectin 3, a pre-mRNA splicing factor in acute inflammatory responses),³⁴ and IL17RA (Interleukin 17 Receptor A, a receptor of IL17 (a proinflammatory cytokine) secreted by activated T-lymphocytes)³⁵ were identified as critical / shared genes among the biological terms. In addition, the top 200 DEGs were also involved in the processes of mRNA processing pathways, metabolic pathways, and kinase activities. Inflammatory response and cell proliferation of CMV-CTLs might be more promptly increased in the CMV-seropositive donor group compared with the CMV-seronegative group.

● ***Comparison according to the CMV-reactivation pattern in the subgroup of CMV-***

seropositive or - negative donor

Since we found that the GEP differed according to the donor CMV serostatus, we then checked the difference in GEP according to the CMV reactivation pattern in the sub-cohorts of donor CMV status. Focusing on the sub-cohort of recipients with CMV seropositive donors (n=10), increased TNF still seemed to centrally work in the no-CMV reactivation group in addition to oxygen carrier activities like HBB (Hemoglobin Subunit Beta, an oxygen transporter) and filament organization such as SNAP25 (Synaptosome Associated Protein 25, a regulator of neurotransmitter release)³⁶ (Fig. S5A-C, DatasetS3c). Focusing on the sub-cohort of recipients with CMV seronegative donors (n=11), genes associated with cell replication /division such as UBE2C (Ubiquitin Conjugating Enzyme E2 C, a member of the E2 ubiquitin-conjugating enzyme family)³⁷ and MAD2L1 (Mitotic Arrest Deficient 2 Like 1, a component of the mitotic spindle assembly checkpoint)³⁸ were increased in the no-CMV reactivation group (Fig. S6, DatasetS3d). On the other hand, CCR2 (C-C Motif Chemokine Receptor 2, a receptor for monocyte chemoattractant protein-1)³⁰ and IL7R (Interleukin 7 Receptor, a receptor for IL7) were increased in the repeated CMV reactivation group and reduced in the other groups (Fig. S6, DatasetS3d).

Next, we focused only on GO terms of the immune system bioprocesses. In the CMV-seropositive donor cohort, GO enrichment analyses using the top 500 DEGs revealed that they were involved in “T-cell-mediated immunity” including cytokine production, “T-cell costimulation”, “negative regulation of lymphocyte activation”, and “mast cell-mediated immunity” (Fig. 8A). The shared genes associated with cytokine production such as TNF, IL1R1 (Interleukin 1 Receptor Type 1, a receptor of an important mediator, IL1, involved in many cytokine-induced immune and inflammatory responses)³⁹ and RSAD2 (Radical S-Adenosyl Methionine Domain Containing 2, an interferon-inducible antiviral protein that belongs to the S-adenosyl-L-methionine (SAM) superfamily of enzymes)⁴⁰ were increased in the no-CMV reactivation group (Fig. 8B). In addition, the genes associated with negative regulation of T-cell activation such as LAX1 (Lymphocyte Transmembrane Adaptor 1, a negative regulator of TCR-mediated signalling in T-cells and BCR (B-cell antigen receptor)-mediated signalling in B-cells),⁴¹ TNFRSF21 (TNF Receptor Superfamily Member 21, a negative regulator of T-cell responses triggered by TCR stimulation),⁴²

and DLG5 (Discs Large MAGUK Scaffold Protein 5, a regulator of the Hippo signalling pathway involved in regulating cell proliferation)⁴³ were also increased in the no-CMV reactivation group (Fig. 8B). On the other hand, LILRB2 (Leukocyte Immunoglobulin Like Receptor B2, a member of the leukocyte immunoglobulin-like receptor family, transducing a negative signals and inhibiting stimulation of an immune response)⁴⁴ was increased in the repeated CMV reactivation group (Fig 8B). In the CMV-seronegative donor cohort, the GO enrichment analyses using the top 324 DEGs with a false discovery rate (FDR) <0.4 revealed that they were involved in immunological bioprocesses of “lymphocyte proliferation”, “regulation of T-cell activation”, and “regulation of myeloid cell differentiation” (Fig. 8C). The lymphocyte proliferation process included not only the regulation of T-cell but also the regulation of B-cell and macrophage chemotaxis. The genes associated with T-cell costimulation such as CD160 (CD160 Molecule, the expression is closely associated with peripheral blood NK cells and CD8 T lymphocytes with cytolytic effector activity),⁴⁵ CD80 (CD80 Molecule, a membrane receptor that is activated by the binding of CD28 or CTLA-4)⁴⁶ and EPHB6 (EPH Receptor B6, a kinase-defective receptor for members of the ephrin-B family that inhibits JNK activation and TCR-induced IL-2 secretion)⁴⁷ were increased in the no-CMV reactivation group (Fig. 8D). On the other hand, LILRB1 (Leukocyte Immunoglobulin Like Receptor B1, a member of the leukocyte immunoglobulin-like receptor family, transducing inhibitory signals and down-regulation of the immune response)⁴⁸ as well as CCR2 and IL7R was increased in the repeated CMV reactivation group (Fig. 8D). In summary, when we focused on the CMV-seropositive donor cohort, cytokine production would be increased, but further or excess T-cell activation would be suppressed in the no-CMV reactivation group. On the other hand, when we focused on the CMV-seronegative donor cohort, T-cell costimulation signalling would be increased in the no-CMV reactivation group, while negative regulation of T-cell activation / proliferation might still be increased in the repeated CMV reactivation group.

GEP, PPI and GO enrichment analyses of HLA-A24-restricted CMVpp65-specific CTLs in the late phases after allo-HCT

● ***Comparison according to the CMV-reactivation pattern***

The top 200 DEGs are shown in Dataset S4a. In the clustering heatmap by the top 100 DEGs, they did not seem to be clearly divided according to the CMV reactivation pattern compared with that in the early phase (Fig. 9A). The PPI network demonstrated that angiogenesis-related genes like VEGFA (Vascular Endothelial Growth Factor A, a growth factor active in angiogenesis, vasculogenesis and endothelial cell growth), immune response-related genes like FCGR2A (Fc Fragment Of IgG Receptor IIa, a low affinity receptor for immunoglobulin gamma, promoting phagocytosis and cellular responses against pathogens),⁴⁸ and cell replication-related genes like MAD2L1 seemed to work as hub genes in the network (Fig. 9B). Focusing on the GO terms of immune systems, the top 200 DEGs were involved in the processes of $\alpha\beta$ T-cell activation, NK cell immunity, and leukocyte cytotoxicity (Fig. 9C). The shared genes such as TRAC (T Cell Receptor Alpha Constant, a constant region of TCR α chain), TUBB (Tubulin Beta Class I, a beta tubulin protein), IL18 (Interleukin 18, A proinflammatory cytokine primarily involved in polarized T-helper 1 cell and NK cell immune responses),⁴⁹ and IL6R (Interleukin 6 Receptor, a subunit of the interleukin 6 (IL6) receptor complex)⁵⁰ for the immunological processes remained increased in the repeated CMV reactivation group (Fig. 9D), suggesting that CMV-CTLs might still have to work actively in the repeated CMV reactivation group even in the late phase after allo-HCT and to recruit other immune cells, while those in the other two groups would be under a steady state without CMV reactivation for a long time. Alternatively, the CMV-CTLs themselves may be ineffective against the host CMV but be capable of signalling NK cells, B cells and components of the innate immune system.

● ***Comparison according to donor CMV-serostatus***

The top 200 DEGs are shown in Dataset S4b. There was only one gene with FDR <0.05 and three genes with FDR <0.4, suggesting that there was no robust difference in GEP in the comparison in the late phase after allo-HCT. Therefore, PPI and GO enrichment analyses were not performed.”

In the Discussion section:

“As expected, inflammatory cytokines like TNF were increased in CMV-CTLs of the no-CMV reactivation group in the early phase of allo-HCT. According to the clone dominance observed between a donor and recipient post allo-HCT in the no-CMV reactivation group with CMV-positive donors as mentioned above (Fig. S4C), the immunity transferred from CMV seropositive donors was considered to mainly keep playing a role against CMV throughout allo-HCT in this group. On the other hand, the immunity against CMV should be naively introduced after allo-HCT from CMV-seronegative donors. The CMV-CTLs transferred from CMV-seropositive donors are generally considered to be “older”, while CMV-CTLs naively-introduced from CMV-negative donors after allo-HCT should be “younger”. Thus, we initially expected that the CMV-CTLs in the CMV-seropositive donor group might have a lower expression of CD28 (a costimulatory receptor for TCR signals) since reduction or loss of CD28 is known to be an indicator of cell senescence or exhaustion.^{59,60} However, we failed to detect the difference in CD28 expression between the CMV-seropositive and -negative donor groups. Therefore, CMV-CTLs in the CMV-seropositive donor group may include functional stem-memory CMV-CTLs for efficient self-renewal and smooth transition to effector CTLs as reservoirs of highly functional memory T-cells,^{57,61} and prevent exhaustion of overall CMV-CTLs. If we consider the DEGs according to the CMV reactivation pattern only in the CMV-seropositive donor cohort, the negative regulation of T-cell activation by several genes like LAX1⁴¹ and TNFRSF21⁴² as well as cytokine production pathways were enriched in the no-CMV reactivation group. The negative regulation of T-cell activation might be simultaneously upregulated to prevent the excessive exhaustion of CMV-CTLs. In fact, PD-1 (Programmed Cell Death 1, an immune-inhibitory receptor expressed in activated T cells and an established marker of exhaustion) was not listed as DEG in this study, supporting previous reports that PD-1 expression on CMV-CTLs was lower than that of tumor- or other-virus-specific CTLs and its expression on CMV-CTLs may be independent of T-cell exhaustion.^{59,62,63} When we focused on the CMV-seronegative donor cohort, CD160 and CD80 were increased in the no-CMV reactivation group. There is some debate regarding whether CD160 on T-cells have costimulatory or coinhibitory signals. The expression of CD160 on NK-cells has been reported to trigger cytotoxic activity,⁴⁵ and that on CD8+ T-cells is considered to

costimulate the proliferation of activated T cells,⁶⁴ indicating the costimulatory potential of CD160 signaling. On the other hand, in the setting of chronic viral infection such as HIV, CD160 signaling was reported to induce exhaustion and functional impairment specific to influenza, EBV, and CMV.⁶⁵ Recently, in pancreatic cancer, CD160 expression on CD8 + T cells is reported to have active effector responses but limited activation potential.⁶⁶ Taken together, CD160 might help to control the acute phase of CMV reactivation, but would not be beneficial if the viral stimulation persisted for a long time. CD80 is mainly expressed on antigen-presenting cells, but is also detected on CD8+ T-cells. The activation of CD80 by the binding of CD28 or CTLA-4 induces both stimulatory and inhibitory signalling. Recently, CD80 expression on memory CD8+ T cells after acute viral infections has been reported to play an important role in suppressing excessive CD8+ T cell recall responses, leading to an appropriate recall immune response.⁴⁶ The expression of CD80 might also be favorable for the no-CMV reactivation group. On the other hand, the expression levels of IL7R, CCR2, and LILRB1 were increased in the repeated CMV reactivation group of the CMV-seronegative donor cohort. Since IL7R expression is known to be reduced on activated virus-specific effector T-cells^{59,67} and CCR2 is reported to be downregulated in memory /effector T-cells following TCR stimulation,³⁰ the increased IL7R and CCR2 expression in the repeated CMV reactivation group may suggest a relative lack of efficiently-activated and -proliferated CMV-CTLs compared to those in the other two groups. Interestingly, inhibitory receptors like LILRB1 (also known as CD85j) and LILRB2 (also known as CD85d) were increased in the repeated CMV reactivation group in the CMV-seronegative and -positive donor cohorts, respectively. LILRB1 has been reported to be mainly expressed on the terminally-differentiated effector T-cells and to increase with age,⁴⁸ suggesting a kind of cell senescence. Therefore, repeated CMV reactivation after allo-HCT may promote cell senescence of CMV-CTLs. The inhibition of LILRB1 has been reported to promote /enhance the efficient proliferation of CMV-CTLs.⁴⁸ In addition, both of LILRB1 and LILRB2 have recently been considered as innate and adaptive immune checkpoint molecules.⁶⁹ Targeting these increased receptors might be a candidate of future treatment strategies to recruit of CMV-CTLs for the repeated CMV reactivation group. It would be

warranted to investigate the association of these identified DEGs with efficient CTL function using actual patient samples in the context of allo-HCT and CMV reactivation.”

Specific points:

6) it is called "adoptive" T cell therapy (not adaptive; T cells are adopted by the host organism)

7) in addition to refs 18-21, more important studies showing proof-of-concept of CMV-specific T cell therapy should be mentioned, particularly PMIDs 1352912, 7675046, 16061727, 28090089

- Author response:

Thank you for this suggestion. We modified the word through the text. According to the reviewer's request, we also added the references.

8) Fig. 1 should be moved to the supplementary figures and be mentioned after the general set-up (i.e. how many CTLs from how many samples from how many patients were analyzed etc)

9) More detailed information on the patients needs to be clearly visualized in the first main figure and also be more explicitly explained in the text beyond the mere indication to Dataset S1 (i.e. How many cells were analyzed for each patient? How do the patients distribute across the reactivation groups, i.e. no reactivation, one episode, repeated. How are early phases defined etc. These important information are so far only hidden in the methods and in the supplementary data)

- Author response:

Thank you for this advice. We modified the previous Fig. 1 into the current Fig. S1, and added Table 1 to provide more information across the groups.

In addition, we added a section on patient characteristics to the first paragraph of the Results and provide detailed information as follows:

“Patient characteristics

We analyzed TCR of CMV-CTLs in 51 samples obtained from 26 patients and 3 donors (Table 1), including 2 samples analyzed by the direct single-cell method alone and reported previously^{9,10}: 26 samples in the early phase (1 to 3 months after allo-HCT), and 22 samples in the late phase (6 months to >1 year after allo-HCT). All of the recipients and their corresponding donors had to have HLA-A24:02 or -A24:20. The median ages of the recipients and donors were 45 (range:16-4) and 37 years (range:11-53), respectively. All recipients tested positive for CMV, while half of the corresponding donors tested negative for CMV. Of them, CMV reactivation was not observed in 10 (the no-CMV reactivation group). Only one episode of CMV reactivation was observed in 9 (the one-episode group), while 7 recipients experienced repeated CMV reactivation (the repeated CMV reactivation group). As expected, the CMV reactivation group frequently experienced grade 2-4 acute graft-versus-host disease (Table1).”

Furthermore, actually sorted cell counts for each patient are shown in Dataset S1, and we show how many cells were analyzed in the first paragraph of the Results section, under “Features of TCR β -CDR3 in HLA-A24-restricted CMVpp65-specific CTL clones”:

“In total, TCR β -CDR3 were analyzed in 19765 CMV-CTLs from the 51 samples [median: 215 cells (range: 30-1000) /sample], and 354 clones were identified (Dataset S1). Of the clones, 346 were observed after allo-HCT and the remaining 8 clones were observed only in the donors.”

10) Fig. 2a: Why is BV7, if it is the most frequent TRBV chain observed, not displayed in the 3D plot? The same occurs for BJ2-1 and 2-3. In general, the 3D plot does not seem to convey any additional help for interpretation beyond the Circos plot in Fig. 2b and should be therefore removed. d.

- Author response:

Thank you for this advice. Accordingly, we deleted the 3D plots, and changed the

numbers of subsequent figures.

11) Fig. 2d: In order to be better able to evaluate the data, the number of clones and number of patients that contributed to each plot should be indicated in the figure legend; the same applies also to other sub-figures, e.g. Fig. 3 in which it is not clear what $n=x$ is referring to. In general, the manuscript lacks specific information on many occasions, for example also in line 123 (major clones accounting for $\geq 5\%$ of what?)

- Author response:

According to the reviewer's request, we added the actual counts of the analyzed clones to the figures and modified the corresponding legends (Fig. 1) as mentioned above.

In addition, we added the definition of major clones to the corresponding figure legend (Fig. 2) as mentioned above.

Similarly, we modified the definition throughout the text as follows:

In the 4th paragraph on pg.6: "Next, we focused only on the major CMV-CTL clones that accounted for $>5\%$ of all CMV-CTLs among individual recipients in the early or late phases after allo-HCT."

In the last paragraph on pg.7: "When we focused on the major clones that accounted for $>5\%$ of all CMV-CTLs among individual recipients in the early or late phases after allo-HCT".

12) Lines 119-122: Why do the authors hypothesize that clones with GGG in ≤ 15 AA or GG in ≥ 16 AA or clones in the no-CMV reactivation group show a high binding affinity? pp65 is rather expressed in the acute phase of the infection (and not during latency, for which a reverse repertoire evolution towards dominance of low affinity clones has been recently shown, PMID 32205883). So, I would hypothesize that after repeated infection, avidity maturation should take place, with preferential selection of high affinity clones. This actually seems to be supported by the data (Fig. 3F), even though the differences seem to be not statistically significant. Whether clones with GGG in ≤ 15

AA or GG in ≥ 16 AA would show high or low affinity I would not have been able to predict. I understand that those were features that were disproportionately often found in the CTLs. However, they were not functionally selected after repeated infections (see argument above). In general, binding predictions based on CDR3 beta alone are only a weak indicator of TCR affinity. First, TCR affinity is dependent on both CDR3 beta and alpha. Second, to my knowledge no experimental/functional data exist that show that TCR binding predictions are able to predict TCR affinity well.

- Author response:

Thank you for this advice. Initially, we simply thought that dominant clones in the no-CMV reactivation group might have tended to be easy to bind and have worked functionally, because they did not suffer from CMV despite immunosuppressive conditions following allo-HCT. We understand the reviewer's opinion that after repeated infection, avidity maturation might take place with preferential selection of high-affinity clones, when we look at the level of individual clones (current Fig. 2E). Thus, for the purpose of checking the effect as a mass, we weighted their binding scores by the cell counts of individual clones. The weighted values were higher in the no-reactivation group in the early phase after allo-HCT among the major clones (current Fig. 2F). Taken together, these findings show that clones with higher affinity also appeared in the repeated reactivation groups, but the entire effect as a clone cohort might be higher in the no-CMV reactivation group. The additional analyses were added to the Results section on pg.8 as follows:

“Clones with a higher binding score equally appeared in the repeated CMV reactivation group (Fig. 2E). However, when their binding scores were weighted by the estimated cell counts of individual clones in cryopreserved PBMCs isolated from 10ml of peripheral blood, the weighted values were higher in the no-reactivation group in the early phase after allo-HCT among the major clones ($P= 0.02$ by the Kruskal-Wallis test & $P=0.031$ by the J-T test for decreasing tendency, Fig. 3F). On the other hand, the weighted values were not significantly different according to the CMV reactivation group in the late phase after allo-HCT ($P=0.13$ by the Kruskal-Wallis test).”

Furthermore, we added the following discussion on pg.19 according to the reviewer's opinion and the additional results:

“Notably, the binding scores weighted by the cell counts of individual clones were increased in the no-reactivation group (Fig. 2F), although clones with a higher binding score equally appeared in the repeated CMV reactivation group (Fig. 2E). Thus, CMV reactivation post allo-HCT would be influenced not only by the TCR sequence itself, but also by whether T-cells could properly proliferate and functionally work under immunosuppressive conditions post allo-HCT. Actually, the RNA-sequence of CMV-CTLs suggested that their DEGs according to the CMV reactivation pattern and donor CMV serostatus were involved in the network associated with the effector cytokines like TNF or in that of cell division. Therefore, in addition to the TCR sequence / structure analyses, further investigations of efficient cell proliferation, differentiation, cytokine production, and senescence using actual recipient samples will be required for the optimized control of CMV reactivation by CMV-CTLs.”

We also agree with the reviewer's suggestion that a weak point of our analyses was focusing only on TCR-beta. We added the following sentences to the Limitation section:

“Third, we analyzed only TCR β -CDR3, and not TCR α . In addition, the predicted loop structure of TCR β -CDR3 was not validated by X-ray crystallography. Therefore, the actual binding affinity might be different from the predicted score in each T-cell. Our study was based on observational facts. Functional analyses would be warranted in the future, if all CMV-CTL clones could be established and all pairs of TCR α and TCR β could be linked in individual clones.”

We'd be most grateful if you gave us generous consideration.

13) In this regard, the predicted loop structures of Fig. 3G are aesthetically nice, but their contribution to an enhanced understanding is questionable. How is the statement "seemed to have a relatively symmetric groove" (line 130) supported by the data? What kind of predictions could be made based on the predicted loop structures? Can those predictions

then be tested/validated? This is just one of many examples in the manuscript where qualitative observations are not quantitatively/functionally validated.

- Author response:

Thank you for thi advice. The mention of a symmetric groove was based solely on the appearance. Therefore, we omitted the word “symmetric”, and changed the corresponding sentences in the Results section as follows:

“The predicted loop structure of 15 AA-long TCR β -CDR3 with “GGG” motifs and higher binding scores seemed simple, while that with lower scores seemed more distorted (Fig. 3).”

As the reviewer mentioned, the predicted structure could not be validated, since it was hard to crystallize TCR and to check it by X-ray analyses in our institution. We added the following to the Limitation section, as mentioned above.

“Third, we analyzed only TCR β -CDR3, and not TCR α . In addition, the predicted loop structure of TCR β -CDR3 was not validated by X-ray crystallography. Therefore, the actual binding affinity might be different from the predicted score in each T-cell. Our study was based on observational facts. Functional analyses would be warranted in the future, if all CMV-CTL clones could be established and all pairs of TCR α and TCR β could be linked in individual clones.”

14) Fig. 4 is difficult to understand with the information available. Fig. 4D/E: It is not clear, whether early-late comparisons were done for all reactivation groups pooled. Because if they were, this could average out differences. What would this look like for the individual reactivation groups? Fig. 4F: It is not clear, whether samples (bars in a given order) represent longitudinal samples (early vs. late) in the same patients. I assume each color annotates one clone (this is neither stated in the legend nor in the figure itself). Also, the figure description in the text is not concise et al. In Fig. S3 (which is unfortunately in the supplementary), it is much clearer, because it is clear that these are longitudinal observations, the important information with the clone distribution in both donor and recipient is given, it is clear which reactivation group these data belong to and

which time points are shown. Visualizations like this should be implemented for the other reactivation groups as well. Overall it is not surprising that the repertoires remain stable from early to late in the no reactivation group and that they change upon single or repeated reactivations. What is surprising is that the repertoire diversity is higher in the early phase of the repeated CMV reactivation, because one would expect repertoire focusing to have taken place. Furthermore, potential reasons for a more diverse repertoire at late compared to early time points are not discussed/tested. Could this be due to newly formed T cells originating from the transferred HSCs? The authors should describe and discuss this more clearly, also in the context of the many studies on TCR repertoire diversity during CMV infection that exist.

- Author response:

Thank you for this advice. As mentioned by the reviewer, Fig. 4D/E was made for all of the pooled CMV-CTLs. According to the reviewer's advice, we added several figures on changes in proportion and diversity according to the CMV reactivation pattern (current Fig. S4A-B). We modified the sentences as follows:

“Focusing on time-dependent changes (Fig. 4D), the proportions of CMV-CTLs decreased in the late phases ($P < 0.01$, Fig. 4E) and their diversity increased ($P = 0.046$, Fig. 4F) when all groups were pooled. However, the statistical significance tended to diminish if we checked the time-dependent changes according to the CMV reactivation pattern (Fig. S4A and S4B). If we consider the estimated cell counts, no significant time-dependent changes were observed according to the CMV reactivation pattern.”

Furthermore, according to the reviewer's request, we deleted the previous Fig. 4F. Instead, we added the current Fig. 4D, which shows changes in the clone proportion in each patient between the early and late phases. The previous Fig. 4 G-I has been moved to Fig. S4.

Further modifications were made according to the reviewer's request in major points above.

15) Fig. 5A-C: The figures should be labelled with TSNE-1 and TSNE-2. Also, in my eyes the numbers for the clones refer to clone counts (not proportions, as it is stated in the figure legend). In Fig. 5D-F, if the upper label is supposed to be gene names, it is useless since it is not legible. For the figure to be interpretable, sub-figures D-F should be much larger. These are just few of many examples of sloppy annotations of figures and legends as well as sloppy figure design (legibility, sub-figure sizes, font sizes, font styles etc etc) throughout the whole manuscript.

16) Fig. 6A: It is nice to see how consistently the patients cluster according to the clinical group. However, again, gene names are not legible and the sub-figure should be larger.

- Author response:

Thank you for these comments. We labeled the figures with tSNE-1 and tSNE-2, and changed “proportion” to “counts” in the legend.

In addition, we modified the corresponding section, and deleted Fig. 5D-F, since only TCR genes were basically significant. We show the lists of differentially expressed genes in supplementary documents, and modified the figure legend as mentioned above.

17) Fig. 6A: It is nice to see how consistently the patients cluster according to the clinical group. However, again, gene names are not legible and the sub-figure should be larger.

- Author response:

According to the advice of the reviewer, we modified all figures and made them as legible as possible. In addition, the legends were modified as mentioned above.

18) lines 198-200: references for the individual genes should be included; also, it is not clear how " degrees of centrality and read counts" (line 197) were quantitatively used as cut-offs / filters.

- Author response:

Thank you for this advice. We added an explanation of the identified genes, a functional summary, and references for individual genes.

In addition, we explained the degrees of centrality as follows in the first paragraph of the Results section on GEP and in individual figure legends: “the degrees of centrality, meaning how many edges are connected to each gene node”

19) headers in lines 191, 205 and 214 have (accidentally?) circles before the headers
20) Fig. 7, S4 and S5: See comments above regarding figure design: Illegible gene names, sloppy annotations, incomplete legends (e.g. degree of what? in Fig. 7, although this is mentioned in the legends). Importantly, does a high \log_2fc indicate upregulation in CMV-positive individuals? A discussion/interpretation or at least an attempt of contextualization of these findings is missing in the manuscript.

- Author response:

Thank you for this advice. We completely modified the figures and legends. In addition, we discuss them as mentioned above.

We'd be most grateful if you gave us generous consideration.

To Reviewer #2:

We believe that there were no comments from Reviewer #2. We appreciate that the reviewer might find our paper suitable for publication.

To Reviewer #3:

Reviewer#3's comments:

1) Table 1 is instructive but raises concerns regarding the results presented unless certain data regarding the patients is integrated into the data presented in the figures. This is particularly the case for the data presented regarding the TCRs and T-cells derived from patients at specific times post transplant. Specifically:

- a. 9 of the 26 patients received transplants from HLA disparate donors. Did all recipients and donors share HLA 2402 and A 2420?
- b. Was the donor's, or the post transplant recipient's T-cell response to the QYD peptide presented by HLA 2402 the dominant response?

The concern here is that if the recipient does not share HLA 2402, the donor's T-cells would not recognize the QYD peptide and would fall off in frequency. Similarly, if the donor inherits HLA B0702 which is always dominant, the proportion of tetramer+ cells would be reduced. This could contribute to the lower proportion of Tet+ cells in patients with recurrent CMV viremia.

- Author response:

Thank you for this question. All recipients and donors shared HLA-A2402 or -A2420. Therefore, the QYD peptide was thought to be also recognized by donor T-cells. Only one recipient/donor pair had HLA-B0702, and their samples were analyzed only in the early phase post allo-HCT. Unfortunately, we could not check the T-cell response for other peptides or HLAs, since the samples from actual patients were quite limited. However, we fully agree that the comparison of T-cell responses among other HLAs sounds very interesting, and would like to study it in the future.

According to the reviewer's advice, we added the following sentence to the section on study design:

"All of the recipients and their corresponding donors had to have HLA-A24:02 or -A24:20."

In addition, we added the following explanation of HLA-mismatch to the legend of

Table 1:

“All of the recipients and their corresponding donors shared HLA-A24:02 or -A24:20.”

Furthermore, we added the following sentence to the Limitation section of the Discussion: “Second, we focused on only HLA-A24-restricted CMV-CTLs. Other class-I HLA-restricted CMV-CTLs could also play a role against CMV, which might have contributed to the lower proportion of tetramer-positive cells in patients with recurrent CMV viremia. We must assess the overall repertoire diversity beyond HLA restriction to clarify clinical aspects in the future.”

2) Were the descriptions of the TCR sequences and their binding affinities in patients early and late post transplant based only on patients who received transplants from seropositive donors? If this is the case, it would be useful to state this up front, it would greatly clarify interpretation of data. If T-cells generated post transplant from seronegative donors are also included, were their TCR sequences shorter, or lacking in GGG sequences, or of lower affinity?

- Author response:

Thank you for the great suggestion. Binding scores were calculated for CMV-CTLs derived from both CMV-seropositive and -seronegative donors. We checked the difference between the CMV-seropositive- and -seronegative donor groups, and found that the TCR sequences in the seronegative donors tended to have lower scores than those of the seropositive donors ($P=0.06$), although there was no significant difference in AA length of TCR ($P=0.22$) or the presence of the “GGG” motif among the major clones.

According to the reviewer’s suggestion, we added the following sentences to page 7:

“There was no difference in AA length of TCR ($P=0.22$) or the presence of the “GGG” motif between CMV-seropositive and -seronegative donors ($P=0.56$).”

In the section on the TCR-peptide binding score, we added “The binding score

tended to be higher in the CMV-seropositive donor groups ($P=0.06$)”.

3) The data presented in Figure 3 provide evidence that the CDR3 amino acid sequences for patients who don't develop CMV reactivations are longer, and are more likely to contain GGG motifs. Those longer TCRs with GGG motifs are also of higher affinity with more symmetrical structures potentially better able to bind peptide HLA complexes. These data also support the central hypothesis of the next section which is that patients who don't experience a CMV reactivation, or only have 1 such reactivation have received transplants from seropositive donors whose dominant CMVpp65 specific T-cells are better able to confer resistance on the transplanted host. Against this hypothesis, however, is Figure 3F, which suggests that CMVpp65 T-cells that are A24 restricted and have TCRs that exhibit GGG motifs and presumably higher affinity are present at frequencies that are not different in both the no reactivation and the multiple CMV viremia episode groups. The authors provide no explanation for Figure 3F, bringing into question whether any of the data on the TCRs has clinical significance. It is also not considered in the Discussion. However, there are many possibilities that could explain Figure 3F, both in terms of the T-cells (e.g. actual number of the T-cells in the blood, level of expression of costimulatory molecules; effector cytokines such as TNF or IFN γ , or the T-cell cytotoxic activity) or properties of the patient's endogenous virus (e.g. evasins, other factors preventing antigen presentation). This needs to be addressed.

- Author response:

We appreciate this great suggestion. We had expected to see lower affinity in the repeated CMV reactivation group, but similar affinity distributions were unexpectedly observed according to the reactivation patterns. One possible explanation is that the comparison of the binding score among CMV reactivation patterns was based on the TCR itself, and the cell counts of individual clones were not considered, as mentioned by the reviewer. Thus, we weighted the binding scores with the cell counts of individual clones, and found that the weighted binding scores of major clones in the early phase of allo-HCT were significantly increased in the

no-activation group. We added a figure of the weighted scores in Fig. 2F. In addition, we added the following sentences to the corresponding section in the Results on pg.8:

“Clones with a higher binding score equally appeared in the repeated CMV reactivation group (Fig. 2E). However, when their binding scores were weighted by the estimated cell counts of individual clones in cryopreserved PBMCs isolated from 10ml of peripheral blood, the weighted values were higher in the no-reactivation group in the early phase after allo-HCT among the major clones (P=0.02 by the Kruskal-Wallis test & P=0.031 by the J-T test for decreasing tendency, Fig. 3F). On the other hand, the weighted values were not significantly different according to the CMV reactivation group in the late phase after allo-HCT (P=0.13 by the Kruskal-Wallis test).”

Furthermore, we added the following discussion:

“Notably, the binding scores weighted by the cell counts of individual clones were increased in the no-reactivation group (Fig. 2F), although clones with a higher binding score equally appeared in the repeated CMV reactivation group (Fig. 2E). Thus, CMV reactivation post allo-HCT would be influenced not only by the TCR sequence itself, but also by whether T-cells could properly proliferate and functionally work under immunosuppressive conditions post allo-HCT. Actually, the RNA-sequence of CMV-CTLs suggested that their DEGs according to the CMV reactivation pattern and donor CMV serostatus were involved in the network associated with the effector cytokines like TNF or in that of cell division. Therefore, in addition to the TCR sequence / structure analyses, further investigations on efficient cell proliferation, differentiation, cytokine production, and senescence using actual recipient samples will be required for the optimized control of CMV reactivation by CMV-CTLs.”

4) The section on the changes in proportion and diversity of the A24 restricted CMVpp65-specific T-cells is of interest in that it provides evidence that dominant CMV T-cells in the donor are also dominant in the host. The data also suggest that there

is the expected diversification of T-cells late post HCT, but that while the dominant clone usually persists, it falls off in patients with recurrent infections. Again, the data presented only address proportions of cells, no actual numbers. This needs to be clarified since the number of antigen-specific T-cells is essential to the response. Also, neither the results section nor the Discussion provide the author's thoughts as to why the dominant clones are lost, nor an explanation for the absence of a proportionate rise in new clones to address recurrent infection.

- Author response:

Thank you for this suggestion. As mentioned by the reviewer, while the dominant clone usually persisted, it fell off in several cases. Please note that this phenomenon was not limited to the repeated CMV reactivation group. According to the reviewer's suggestion, we additionally analyzed the difference in TCR-binding score between the increased and decreased clones in the late phases, focusing on the dominant clones. We found that the increased clones tended to have higher TCR-binding scores than the decreased clones. Clones with a higher binding score might be preferentially selected or efficiently proliferate during the time-course post allo-HCT.

Thus, we modified the section as follows;

“When we considered the estimated cell counts of CMV-CTLs in 10ml of peripheral blood, the median cell count in the repeated CMV reactivation group was significantly lower than those in the other groups: 153 cells (range: 86 - 1030 cells) vs. 1760 cells (range: 60 - 3000 cells) in the one-episode group vs. 880 cells (range: 142 - 3000 cells) in the no-CMV reactivation group, $P=0.046$), suggesting that the actual CMV-CTL counts would be critical for CMV control after allo-HCT. Focusing on time-dependent changes (Fig. 4D), the proportions of CMV-CTLs decreased in the late phases ($P<0.01$, Fig 4E) and their diversity increased ($P=0.046$, Fig. 4F) when all groups were pooled. However, statistical significance tended to diminish if we checked the time-dependent changes according to the CMV reactivation pattern (Fig. S4A and S4B). If we consider the estimated cell counts, there were no significant time-dependent changes according to the CMV reactivation pattern.

If we focus on the three pairs of donor and recipient in the no-CMV reactivation group transplanted from CMV-seropositive donors, the dominant clones of each donor remained dominant after allo-HCT (Fig. S4C), suggesting that CMV-CTLs transferred from the donors kept playing an anti-viral role in the cohort. Similarly, if we considered the most dominant clone of individual patients in all groups, there was no significant time-dependent proportional change according to the CMV-reactivation pattern between the early and late phases after allo-HCT (Fig. S4D). This meant that the dominant clone in the early phases after allo-HCT usually remained dominant. However, in several cases, some dominant clones decreased in the late phases after allo-HCT (Fig. 4D). Next, we focused on the dominant clones that accounted for > 35 % of all CMV-CTLs within individual recipients in the early or late phases after allo-HCT, and checked the difference in TCR-binding scores between the decreasing and increasing clones in the late phases. The binding scores of the increasing dominant clones in the late phases were significantly higher than those of the decreasing scores (P=0.016, Fig. 4G)

The top 1 and 2 clones in the early phase of allo-HCT accounted for >75% in all of the no-CMV reactivation group, 78% of the one-episode group, and 43% of the repeated CMV reactivation group (P=0.018 by Fisher's exact test). Those in the late phase accounted for >75% in 78% of the no-CMV reactivation group and 71% of the one-episode group, but in only 33 % of the repeated CMV reactivation group, albeit this difference was not significant (P=0.21).”

According to the reviewer's request, we added the following summary of the section on pg.10: “In summary, focusing on the dominant clones, the proportion of clones with higher TCR-binding scores tended to increase at the late phase, and conversely those with lower scores tended to decrease”.

5) The data presented on the gene expansion profiles of the T-cells are a major and novel feature of the report. The data comparing these profiles in CMV T-cells generated from naïve donors vs seropositive donors are particularly important. Although the authors have previously presented data regarding recipients of grafts from CMV

seropositive donors who experience CMV reactivation, its absence from this report as a frame of reference limits comparative analysis of the gene expression profiles of the T-cells generated from naïve seronegative donors.

The data presented on the profiles late post transplant are of particular interest, and deserve further discussion, since the activation of genes for CD28 and other costimulatory molecules in the No-P group may reflect their key role in enhancing the effector function of lower avidity T-cells. Similarly, the genes activated in the T-cells from individuals with repeated episodes of CMV viremia suggest that the T-cells themselves may be ineffective against the host CMV but are capable of signaling NK cells, B cells and components of the innate immune system.

- Author response:

We are pleased to hear the reviewer found our analyses important. According to the reviewer's request, we additionally analyzed samples of the recipients who had received graft from CMV-seropositive donors and experienced reactivation. In addition, we re-analyzed all RNA-seq data, and the mapping, annotation, and calculation of gene expression were re-performed with DRAGEN using GRCh38.primary_assembly.genome, instead of the previous platform using Genedata Profiler Genome, STAR, and GRCh37. Thankfully, the additional analyses enabled us to simplify the comparisons, and accordingly we modified the Results as follows:

“GEP, protein-protein interaction (PPI) network, and gene ontology (GO) enrichment analyses of HLA-A24-restricted CMVpp65-specific CTLs in the early phase

● ***Comparison according to the CMV-reactivation pattern***

The explanation of individual genes described below was based on GeneCards (The Human Gene Database, <https://www.genecards.org/>). The top 200 differentially expressed genes (DEGs) with $P < 0.05$ are shown in Dataset S3a. They were well-clustered by the top 100 DEGs (Fig. 6A). Their PPI network was constructed using the top 100 DEGs (Fig.6B). According to their degrees of centrality, meaning how many edges are connected to each gene node, TNF (Tumor Necrosis Factor, a

multifunctional proinflammatory cytokine) and HELLS (Helicase, Lymphoid Specific, a lymphoid specific helicase and regulator for the expansion or survival of lymphoid cells)²⁷ seemed to centrally work as hub genes in the network, as well as DNA repair proteins like RAD51AP1 (RAD51 Associated Protein 1, a structure-specific DNA-binding protein involved in DNA repair)²⁸ (Fig. 6B). In addition, TNF was further increased in both the no-CMV reactivation (n=8) and one-episode (n=7) groups compared with the repeated CMV reactivation group (n=6) (Fig. 6B). The GO enrichment analysis demonstrated that the top 200 DEGs were involved in the “regulation of acute inflammatory response” and “regulation of viral genome replication” as an immunological response (Fig. 6C). In these processes, inflammatory genes such as TNF, and ISG20 (Interferon Stimulated Exonuclease Gene 20, an interferon-induced antiviral exoribonuclease)²⁹ were shared among the biological terms. In addition, they were also involved in the processes of transforming growth factor (TGF)- β pathways, regulation of cell division, and other metabolic activities (Fig. 6C). In summary, TNF expression was increased in the no-CMV reactivation group, and cell replication /division would be more active, leading to the proliferation of CMV-CTLs in that group.

● ***Comparison according to donor CMV-serostatus***

The top 200 DEGs are shown in Dataset S3b. They were also well-clustered by the top 100 DEGs (Fig. 7A). The PPI network suggested that the increased genes of CCR5 (C-C Motif Chemokine Receptor 5, a regulator of granulocytic lineage proliferation),³⁰ LY96 (Lymphocyte Antigen 96, a protein that is associated with toll-like receptor 4 on the cell surface and confers responsiveness to lipopolysaccharide),³¹ S100A8 and S100A9 (S100 Calcium Binding Protein A8 and A9, regulators of inflammatory processes and immune response)³² centrally worked in the CMV-seropositive donor group (n=10) (Fig. 7B). On the other hand, RNA helicases like DDX5 (DEAD-Box Helicase 5, an RNA helicase)³³ were increased in the CMV-seronegative donor group (n=11), suggesting that mRNA processing was more active in this group. The top 200 DEGs were involved in “cellular defense response”, “positive regulation of lymphocyte proliferation”, and “production of molecular mediator involved in inflammatory response” as immunological processes (Fig. 7C).

In these processes, inflammatory genes such as CCR5, LY96, LGALS3 (Galectin 3, a pre-mRNA splicing factor in acute inflammatory responses),³⁴ and IL17RA (Interleukin 17 Receptor A, a receptor of IL17 (a proinflammatory cytokine) secreted by activated T-lymphocytes)³⁵ were identified as critical / shared genes among the biological terms. In addition, the top 200 DEGs were also involved in the processes of mRNA processing pathways, metabolic pathways, and kinase activities. Inflammatory response and cell proliferation of CMV-CTLs might be more promptly increased in the CMV-seropositive donor group compared with the CMV-seronegative group.

● ***Comparison according to the CMV-reactivation pattern in the subgroup of CMV-seropositive or - negative donor***

Since we found that the GEP differed according to the donor CMV serostatus, we then checked the difference in GEP according to the CMV reactivation pattern in the sub-cohorts of donor CMV status. Focusing on the sub-cohort of recipients with CMV seropositive donors (n=10), increased TNF still seemed to centrally work in the no-CMV reactivation group in addition to oxygen carrier activities like HBB (Hemoglobin Subunit Beta, an oxygen transporter) and filament organization such as SNAP25 (Synaptosome Associated Protein 25, a regulator of neurotransmitter release)³⁶ (Fig. S5A-C, DatasetS3c). Focusing on the sub-cohort of recipients with CMV seronegative donors (n=11), genes associated with cell replication /division such as UBE2C (Ubiquitin Conjugating Enzyme E2 C, a member of the E2 ubiquitin-conjugating enzyme family)³⁷ and MAD2L1 (Mitotic Arrest Deficient 2 Like 1, a component of the mitotic spindle assembly checkpoint)³⁸ were increased in the no-CMV reactivation group (Fig. S6, DatasetS3d). On the other hand, CCR2 (C-C Motif Chemokine Receptor 2, a receptor for monocyte chemoattractant protein-1)³⁰ and IL7R (Interleukin 7 Receptor, a receptor for IL7) were increased in the repeated CMV reactivation group and reduced in the other groups (Fig. S6, DatasetS3d).

Next, we focused only on GO terms of the immune system bioprocesses. In the CMV-seropositive donor cohort, GO enrichment analyses using the top 500 DEGs revealed that they were involved in “T-cell-mediated immunity” including cytokine production, “T-cell costimulation”, “negative regulation of lymphocyte activation”, and “mast

cell-mediated immunity” (Fig. 8A). The shared genes associated with cytokine production such as TNF, IL1R1 (Interleukin 1 Receptor Type 1, a receptor of an important mediator, IL1, involved in many cytokine-induced immune and inflammatory responses)³⁹ and RSAD2 (Radical S-Adenosyl Methionine Domain Containing 2, an interferon-inducible antiviral protein that belongs to the S-adenosyl-L-methionine (SAM) superfamily of enzymes)⁴⁰ were increased in the no-CMV reactivation group (Fig. 8B). In addition, the genes associated with negative regulation of T-cell activation such as LAX1 (Lymphocyte Transmembrane Adaptor 1, a negative regulator of TCR-mediated signalling in T-cells and BCR (B-cell antigen receptor)-mediated signalling in B-cells),⁴¹ TNFRSF21 (TNF Receptor Superfamily Member 21, a negative regulator of T-cell responses triggered by TCR stimulation),⁴² and DLG5 (Discs Large MAGUK Scaffold Protein 5, a regulator of the Hippo signalling pathway involved in regulating cell proliferation)⁴³ were also increased in the no-CMV reactivation group (Fig. 8B). On the other hand, LILRB2 (Leukocyte Immunoglobulin Like Receptor B2, a member of the leukocyte immunoglobulin-like receptor family, transducing a negative signals and inhibiting stimulation of an immune response)⁴⁴ was increased in the repeated CMV reactivation group (Fig 8B). In the CMV-seronegative donor cohort, the GO enrichment analyses using the top 324 DEGs with a false discovery rate (FDR) <0.4 revealed that they were involved in immunological bioprocesses of “lymphocyte proliferation”, “regulation of T-cell activation”, and “regulation of myeloid cell differentiation” (Fig. 8C). The lymphocyte proliferation process included not only the regulation of T-cell but also the regulation of B-cell and macrophage chemotaxis. The genes associated with T-cell costimulation such as CD160 (CD160 Molecule, the expression is closely associated with peripheral blood NK cells and CD8 T lymphocytes with cytolytic effector activity),⁴⁵ CD80 (CD80 Molecule, a membrane receptor that is activated by the binding of CD28 or CTLA-4)⁴⁶ and EPHB6 (EPH Receptor B6, a kinase-defective receptor for members of the ephrin-B family that inhibits JNK activation and TCR-induced IL-2 secretion)⁴⁷ were increased in the no-CMV reactivation group (Fig. 8D). On the other hand, LILRB1 (Leukocyte Immunoglobulin Like Receptor B1, a member of the leukocyte immunoglobulin-like receptor family, transducing inhibitory signals and down-regulation of the immune response)⁴⁸ as well as CCR2

and IL7R was increased in the repeated CMV reactivation group (Fig. 8D).

In summary, when we focused on the CMV-seropositive donor cohort, cytokine production would be increased, but further or excess T-cell activation would be suppressed in the no-CMV reactivation group. On the other hand, when we focused on the CMV-seronegative donor cohort, T-cell costimulation signalling would be increased in the no-CMV reactivation group, while negative regulation of T-cell activation / proliferation might still be increased in the repeated CMV reactivation group.

GEP, PPI and GO enrichment analyses of HLA-A24-restricted CMVpp65-specific CTLs in the late phases after allo-HCT

● ***Comparison according to the CMV-reactivation pattern***

The top 200 DEGs are shown in Dataset S4a. In the clustering heatmap by the top100 DEGs, they did not seem to be clearly divided according to the CMV reactivation pattern compared with that in the early phase (Fig. 9A). The PPI network demonstrated that angiogenesis-related genes like VEGFA (Vascular Endothelial Growth Factor A, a growth factor active in angiogenesis, vasculogenesis and endothelial cell growth), immune response-related genes like FCGR2A (Fc Fragment Of IgG Receptor IIa, a low affinity receptor for immunoglobulin gamma, promoting phagocytosis and cellular responses against pathogens),⁴⁸ and cell replication-related genes like MAD2L1 seemed to work as hub genes in the network (Fig. 9B). Focusing on the GO terms of immune systems, the top 200 DEGs were involved in the processes of $\alpha\beta$ T-cell activation, NK cell immunity, and leukocyte cytotoxicity (Fig. 9C). The shared genes such as TRAC (T Cell Receptor Alpha Constant, a constant region of TCR α chain), TUBB (Tubulin Beta Class I, a beta tubulin protein), IL18 (Interleukin 18, A proinflammatory cytokine primarily involved in polarized T-helper 1 cell and NK cell immune responses),⁴⁹ and IL6R (Interleukin 6 Receptor, a subunit of the interleukin 6 (IL6) receptor complex)⁵⁰ for the immunological processes remained increased in the repeated CMV reactivation group (Fig. 9D), suggesting that CMV-CTLs might still have to work actively in the repeated CMV reactivation group even in the late phase after allo-HCT and to recruit other immune cells, while those

in the other two groups would be under a steady state without CMV reactivation for a long time. Alternatively, the CMV-CTLs themselves may be ineffective against the host CMV but be capable of signalling NK cells, B cells and components of the innate immune system.

● ***Comparison according to donor CMV-serostatus***

The top 200 DEGs are shown in Dataset S4b. There was only one gene with FDR <0.05 and three genes with FDR <0.4, suggesting that there was no robust difference in GEP in the comparison in the late phase after allo-HCT. Therefore, PPI and GO enrichment analyses were not performed.”

6) The Discussion needs significant revision. As it is, it reiterates the findings in the results but does not put those data in context for comparison with data presented in prior reports. The discussion of the potential significance of CD28 as a survival signal is worthwhile. However, the contribution of CD28 and other costimulatory factors to the function of T-cells with lower affinity TCRs should also be considered, since the function of T-cells with a diversity of TCR affinity is an important finding in this report. Similarly, the discussion of the genes expressed by T-cells from individuals with recurrent CMV needs to be further developed. There is a wealth of information presented here but it needs to be clarified and more extensively explored as a door to the future rather than a fine tuning of the past.

- Author response:

According to this advice of the reviewer, we modified the Discussion as follows:

“As expected, inflammatory cytokines like TNF were increased in CMV-CTLs of the no-CMV reactivation group in the early phase of allo-HCT. According to the clone dominance observed between a donor and recipient post allo-HCT in the no-CMV reactivation group with CMV-positive donors as mentioned above (Fig. S4C), the immunity transferred from CMV seropositive donors was considered to mainly keep playing a role against CMV throughout allo-HCT in this group. On the other hand, the immunity against CMV should be naively introduced after allo-HCT from CMV-

seronegative donors. The CMV-CTLs transferred from CMV-seropositive donors are generally considered to be “older”, while CMV-CTLs naively-introduced from CMV-negative donors after allo-HCT should be “younger”. Thus, we initially expected that the CMV-CTLs in the CMV-seropositive donor group might have a lower expression of CD28 (a costimulatory receptor for TCR signals) since reduction or loss of CD28 is known to be an indicator of cell senescence or exhaustion.^{59,60} However, we failed to detect the difference in CD28 expression between the CMV-seropositive and -negative donor groups. Therefore, CMV-CTLs in the CMV-seropositive donor group may include functional stem-memory CMV-CTLs for efficient self-renewal and smooth transition to effector CTLs as reservoirs of highly functional memory T-cells,^{57,61} and prevent exhaustion of overall CMV-CTLs. If we consider the DEGs according to the CMV reactivation pattern only in the CMV-seropositive donor cohort, the negative regulation of T-cell activation by several genes like LAX1⁴¹ and TNFRSF21⁴² as well as cytokine production pathways were enriched in the no-CMV reactivation group. The negative regulation of T-cell activation might be simultaneously upregulated to prevent the excessive exhaustion of CMV-CTLs. In fact, PD-1 (Programmed Cell Death 1, an immune-inhibitory receptor expressed in activated T cells and an established marker of exhaustion) was not listed as DEG in this study, supporting previous reports that PD-1 expression on CMV-CTLs was lower than that of tumor- or other-virus-specific CTLs and its expression on CMV-CTLs may be independent of T-cell exhaustion.^{59,62,63} When we focused on the CMV-seronegative donor cohort, CD160 and CD80 were increased in the no-CMV reactivation group. There is some debate regarding whether CD160 on T-cells have costimulatory or coinhibitory signals. The expression of CD160 on NK-cells has been reported to trigger cytotoxic activity,⁴⁵ and that on CD8+ T-cells is considered to costimulate the proliferation of activated T cells,⁶⁴ indicating the costimulatory potential of CD160 signaling. On the other hand, in the setting of chronic viral infection such as HIV, CD160 signaling was reported to induce exhaustion and functional impairment specific to influenza, EBV, and CMV.⁶⁵ Recently, in pancreatic cancer, CD160 expression on CD8 + T cells is reported to have active effector responses but limited activation potential.⁶⁶ Taken together, CD160 might help to control the acute phase of CMV reactivation, but would not be beneficial if

the viral stimulation persisted for a long time. CD80 is mainly expressed on antigen-presenting cells, but is also detected on CD8+ T-cells. The activation of CD80 by the binding of CD28 or CTLA-4 induces both stimulatory and inhibitory signalling. Recently, CD80 expression on memory CD8+ T cells after acute viral infections has been reported to play an important role in suppressing excessive CD8+ T cell recall responses, leading to an appropriate recall immune response.⁴⁶ The expression of CD80 might also be favorable for the no-CMV reactivation group. On the other hand, the expression levels of IL7R, CCR2, and LILRB1 were increased in the repeated CMV reactivation group of the CMV-seronegative donor cohort. Since IL7R expression is known to be reduced on activated virus-specific effector T-cells^{59,67} and CCR2 is reported to be downregulated in memory /effector T-cells following TCR stimulation,³⁰ the increased IL7R and CCR2 expression in the repeated CMV reactivation group may suggest a relative lack of efficiently-activated and -proliferated CMV-CTLs compared to those in the other two groups. Interestingly, inhibitory receptors like LILRB1 (also known as CD85j) and LILRB2 (also known as CD85d) were increased in the repeated CMV reactivation group in the CMV-seronegative and -positive donor cohorts, respectively. LILRB1 has been reported to be mainly expressed on the terminally-differentiated effector T-cells and to increase with age,⁴⁸ suggesting a kind of cell senescence. Therefore, repeated CMV reactivation after allo-HCT may promote cell senescence of CMV-CTLs. The inhibition of LILRB1 has been reported to promote /enhance the efficient proliferation of CMV-CTLs.⁴⁸ In addition, both of LILRB1 and LILRB2 have recently been considered as innate and adaptive immune checkpoint molecules.⁶⁹ Targeting these increased receptors might be a candidate of future treatment strategies to recruit of CMV-CTLs for the repeated CMV reactivation group. It would be warranted to investigate the association of these identified DEGs with efficient CTL function using actual patient samples in the context of allo-HCT and CMV reactivation.”

We’d be most grateful if you gave us generous consideration.

Reviewers' comments:

Reviewer #1 (Remarks to the Author):

The authors have improved the manuscript based on the points raised in the initial review stage. While the data could still be presented in a far more concise manner, the data are relevant and provide a valuable resource for the scientific community. However, some remaining major points need to be addressed:

- Fig. 2E/F should be deleted as the meaning of the analyses is questionable
- Fig. 3 should be described in more detail
- For Fig. 5, the EXACT (not estimated) cell numbers should be provided
- The results and discussion section referring to Fig. 6-9 should be condensed to the most important aspects

Reviewer #3 (Remarks to the Author):

This is a massive compilation of data on a series of HLA A2402 and HLA A2420 patients who received allogeneic HCTs from HLA A2402+ matched donors regarding their A2402 restricted CMV T-cells specific for the QMW epitope presented by the HLA A2402 alleles. Much of the data is of interest. The revision contains large additions particularly regarding comparisons between patients who do not have a CMV reactivation vs those with single reactivation vs those with multiple recurrences. Several of the studies conducted are highly informative. In particular, the data in Figure 3 on the AA length and AA sequences, the data in Figure 4C-G and the data in Figures 5 and 6. On the other hand, the data in Figure 3 and Figure 4A and B are unto themselves and, without context, don't add much to the hypotheses the authors propose.

The data on the gene expression profiles is extensive and addresses key points regarding the potential functions of the T-cells in the different clinical groups and in recipients of grafts from seropositive and seronegative donors. Unfortunately, these data are not linked into any functional analyses.

Nevertheless, there is much to study here. However, it would be useful for interpretation to better explain the heat maps in Figure 9A, specifically the differences between the first group of repeats and the second, and the first no reactivation and the second. Are the differences a function of the time post transplant? This is not clear.

The Network diagram in Figures 6-9 are of great interest, but are not easy to interpret. Similarly, the text describing these networks and the genes activated is really hard to read due to the parenthetical additions which name the genes and describe their real or hypothetical functions. It would be better to include a table with the genes grouped and their full names and expected functions for the reader to refer to. Then the gene abbreviations and their links could be more easily seen and the text more easily read.

Unfortunately, there are also several aspects of the study that limit interpretation of the findings described for the T-cells from those who had no CMV reactivation vs those with 1 or multiple reactivations that would affect application to adoptive therapy. Principal among these is the lack of studies of the donor's cells prior to transplant. At issue is whether the findings in the patients with multiple recurrences reflect what the donor had in the beginning of a progressive selection of T-cells in order to achieve control over time. Also not addressed is whether specific attributes of the A2402 restricted T-cells from individuals without CMV reactivation, or with a single reactivation, actually enhance the capacity of these cells to kill or control the growth of autologous cells infected with the virus. The maintenance of dominant clones throughout the course does suggest that these cells have the wherewithal to control infection, provided they are able to expand to numbers capable of exerting such control. This is a logical hypothesis. However, it needs to be demonstrated directly.

Overall, there is a great deal of new information provided in this manuscript. However, in the absence

of data regarding the function of these Tet+ T-cells, the extensive gene expression profiles remain descriptive. However, the comparisons between clinical groups are new, informative, and still valuable. They certainly provide much for thought.

To Reviewer #1

Reviewer's comments:

The authors have improved the manuscript based on the points raised in the initial review stage. While the data could still be presented in a far more concise manner, the data are relevant and provide a valuable resource for the scientific community. However, some remaining major points need to be addressed:

1) Fig. 2E/F should be deleted as the meaning of the analyses is questionable

- Author's response:

Thank you for the suggestion. Since Fig2E and 2F were added after the previous review round according to another reviewer who requested to show the effect of actual sorted-cell counts (but not proportion) on CMV , we'd like to keep them as one of results if possible.

2) Fig. 3 should be described in more detail

- Author's response:

Thank you for the advice. Considering the importance of the previous Fig3 and according to the request from the other reviewer, we decided to delete the previous Fig3.

3) For Fig. 5, the EXACT (not estimated) cell numbers should be provided.

- Author's response:

Thank you for the request. We added the actually-analyzed T-cell counts in the figure legend, and the text as followings:

"The actually-analyzed cell counts with combined information on gene expression and TCR β -CDR3 were 972 (Case 14, Fig. 4A), 747 (Case 4, Fig. 4B), and 501 (Case 3, Fig. 4C) cells after apparently dying cells were deleted."

4) The results and discussion section referring to Fig. 6-9 should be condensed to the most important aspects

- Author's response:

Thank you for the great advice.

We accordingly condensed the sections of the results and discussion for Fig.6-9, by adding Table 2 for gene explanation and only referring genes associated with immune response as far as possible. Thus, the corresponding sections could be shorten with reduction of over one page.

We'd be grateful if you gave us generous consideration.

To Reviewer #3

Reviewer's comments:

1) This is a massive compilation of data on a series of HLA A2402 and HLA A2420 patients who received allogeneic HCTs from HLA A2402+ matched donors regarding their A2402 restricted CMV T-cells specific for the QMW epitope presented by the HLA A2402 alleles. Much of the data is of interest. The revision contains large additions particularly regarding comparisons between patients who do not have a CMV reactivation vs those with single reactivation vs those with multiple recurrences. Several of the studies conducted are highly informative. In particular, the data in Figure 2 on the AA length and AA sequences, the data in Figure 4C-G and the data in Figures 5 and 6. On the other hand, the data in Figure 3 and Figure 4A and B are unto themselves and, without context, don't add much to the hypotheses the authors propose.

- Author's response:

Thank you for the suggestion. According to the request of the reviewer, the previous Figs.4A-C were placed as supplementary documents, and the previous Fig.3 were deleted from the manuscript.

2) The data on the gene expression profiles is extensive and addresses key points regarding the potential functions of the T-cells in the different clinical groups and in recipients of grafts from seropositive and seronegative donors. Unfortunately, these data are not linked into any functional analyses. Nevertheless, there is much to study here. However, it would be useful for interpretation to better explain the heat maps in Figure 9A, specifically the differences between the first group of repeats and the second, and the first no reactivation and the second. Are the differences a function of the time post transplant? This is not clear.

- Author's response:

Thank you for the question. We checked the disease status, GVHD, CMV reactivation at the sampling of the late phase. Regarding the no group, the second group (n=2) experienced subsequent hematological or molecular relapse, while 3 of 4 in the first group suffered from GVHD. Regarding the repeat group, 2 of 3 in the second group still suffered from CMV reactivation and severe GVHD. However, the distinct reason still remains unknown. We added the background in the result as following:

"We checked their CMV reactivation, disease status, and GVHD at the sampling of the late phase. The split subgroup of the no-CMV reactivation group (n=2) experienced subsequent

hematological or molecular relapse. In addition, 2 of 3 recipients in the split subgroup of the repeated-CMV reactivation group still suffered from CMV reactivation and severe GVHD. However, the DEGs were not clearly divided when we considered these clinical backgrounds.”

3) The Network diagram in Figures 6-9 are of great interest, but are not easy to interpret. Similarly, the text describing these networks and the genes activated is really hard to read due to the parenthetical additions which name the genes and describe their real or hypothetical functions. It would be better to include a table with the genes grouped and their full names and expected functions for the reader to refer to. Then the gene abbreviations and their links could be more easily seen and the text more easily read.

- Author’s response:

Thank you for the advice. We added Table 2 for the gene explanations. We accordingly condensed the section of the results and discussion for Fig.6-9, by only referring genes associated with immune response as far as possible. Thus, the corresponding result section could be shorten with reduction of over one page.

Furthermore, we added a short summary for each comparison as followings.

-For comparison according to the CMV-reactivation pattern:

“In summary, TNF expression was increased in the no-CMV reactivation group, and cell replication /division would be more active, leading to the proliferation of CMV-CTLs in that group.”

-For comparison according to donor CMV-serostatus:

“Inflammatory response and cell proliferation of CMV-CTLs might be more promptly increased in the CMV-seropositive donor group compared with the CMV-seronegative group.”

-For comparison according to the CMV-reactivation pattern in the subgroup of CMV-seropositive or -negative donor:

“In summary, when we focused on the CMV-seropositive donor cohort, inflammatory cytokine production such as TNF would be increased, but further or excess T-cell activation would be suppressed by the genes like LAX1 in the no-CMV reactivation group. On the other hand, when we focused on the CMV-seronegative donor cohort, T-cell costimulation signalling of CD160 and CD80 would be increased in the no-CMV reactivation group, while a relative lack of efficiently-activated CMV-CTLs may be suggested by CCR2 and IL7R in the repeated CMV reactivation group.”

-For comparison according to the CMV-reactivation pattern at the late phase:

“, suggesting that CMV-CTLs might still have to work actively in the repeated CMV reactivation group even in the late phase after allo-HCT and to recruit other immune cells, while those in the other two groups would be under a steady state without CMV reactivation for a long time. Alternatively, the CMV-CTLs themselves in the repeated CMV reactivation group may be ineffective against the host CMV but be capable of signalling NK cells, B cells and components of the innate immune system.”

4) Unfortunately, there are also several aspects of the study that limit interpretation of the findings described for the T-cells from those who had no CMV reactivation vs those with 1 or multiple reactivations that would affect application to adoptive therapy. Principal among these is the lack of studies of the donor's cells prior to transplant. At issue is whether the findings in the patients with multiple recurrences reflect what the donor had in the beginning of a progressive selection of T-cells in order to achieve control over time. Also not addressed is whether specific attributes of the A2402 restricted T-cells from individuals without CMV reactivation, or with a single reactivation, actually enhance the capacity of these cells to kill or control the growth of autologous cells infected with the virus. The maintenance of dominant clones throughout the course does suggest that these cells have the wherewithal to control infection, provided they are able to expand to numbers capable of exerting such control. This is a logical hypothesis. However, it needs to be demonstrated directly.

- Author's response:

Thank you for the suggestion. We'd actually have like to analyze more donors. To tell the truth, we frequently performed allo-HCT from unrelated donors in our institution, and pre-transplant analyses of unrelated donors were difficult. In fact, the current cohort included only 6 allo-HCT from CMV-seropositive "related" donors. Unfortunately, the limited samples of them had been available for only 3 pairs in the no-CMV reactivation group (Fig S4F) and all samples of them had been used for repertoire analyses.

In addition, we agree that it would be better to perform the functional analyses and to check whether the dominant clones of CMV-CTLs in the no-CMV reactivation group could overcome the reactivation in the multiple reactivation group. Actually, we tried to establish the individual dominant CMV-CTL clones from a single cell after single-cell sorting using actual samples of post-transplant recipients. However, these cells could not sufficiently proliferate for functional analyses. It might be due to specific in vivo circumstances after allo-HCT such as the status on immunosuppressant. Thus, our future studies will collect the related donor cells and to pursue the methods to establish individual clones directly from actual

donor or recipient samples. We hope to compare the role of dominant CMV-CTL clones in donors between the no- and multiple reactivation groups in the future.

According to the reviewer's suggestion, we added the sentences in the limitation section as followings:

"Our study was based on observational facts. Functional analyses would be warranted in the future, if all CMV-CTL clones could be established and all pairs of TCR α and TCR β could be linked in individual clones. Our future studies need to explore logical hypotheses whether the maintenance of the dominant clones in the no-CMV reactivation group can also actually control CMV reactivation in the repeated group, and whether we can predict a high-risk patient for multiple CMV reactivations by analyzing pre-transplant donor cells."

We'd be most grateful if you gave us generous consideration.

5) Overall, there is a great deal of new information provided in this manuscript. However, in the absence of data regarding the function of these Tet⁺ T-cells, the extensive gene expression profiles remain descriptive. However, the comparisons between clinical groups are new, informative, and still valuable. They certainly provide much for thought.

- Author's response:

We are so pleased to hear that the reviewer thought our paper interesting and informative.

We will keep investigating the T-cell response / functions for our future studies.

REVIEWERS' COMMENTS:

Reviewer #3 (Remarks to the Author):

The authors have address all questions and comments in my review and should be considered for publication.